# Dig out, Dig in! Plant-based diet at the Late Bronze Age copper production site of Prigglitz-Gasteil (Lower Austria) and the relevance of processed foodstuffs for the supply of Alpine Bronze Age miners

**Andreas G. Heiss**[1]*, **Thorsten Jakobitsch**[1,2], **Silvia Wiesinger**[1], **Peter Trebsche**[3]

**1** Austrian Academy of Sciences (OeAW), Austrian Archaeological Institute (OeAI), Wien/Vienna, Austria, **2** University of Natural Resources and Life Sciences, Vienna (BOKU), Institute for Botany, Wien/Vienna, Austria, **3** Department of Archaeologies, University of Innsbruck, Innsbruck, Austria

\* andreas.heiss@oeaw.ac.at

**Data Availability Statement:** All relevant data are within the manuscript and its Electronic Supplementary Materials (ESM). All archaeological

## Abstract

This paper starts from theoretical and methodical considerations about the role of archaeobotanical finds in culinary archaeology, emphasizing the importance of processed cereal preparations as the "missing link" between crop and consumption. These considerations are exemplified by the discussion of abundant new archaeobotanical data from the Late Bronze Age copper mining site of Prigglitz-Gasteil, situated at the easternmost fringe of the Alps. At this site, copper ore mining in opencast mines took place from the 11th until the 9th century BCE (late Urnfield Culture), as well as copper processing (beneficiation, smelting, refining, casting) on artificial terrain terraces. During archaeological excavations from 2010 to 2014, two areas of the site were investigated and sampled for archaeobotanical finds and micro-debris in a high-resolution approach. This paper aims at 1) analysing the food plant spectrum at the mining settlement of Prigglitz-Gasteil basing on charred plant macroremains, 2) investigating producer/consumer aspects of Prigglitz-Gasteil in comparison to the Bronze Age metallurgical sites of Kiechlberg, Klinglberg, and Mauken, and 3) reconstructing the miners' and metallurgists' diets.

Our analyses demonstrate that the plant-based diet of the investigated mining communities reflects the general regional and chronological trends rather than particular preferences of the miners or metallurgists. The lack of chaff, combined with a high occurrence of processed food, suggests that the miners at Prigglitz-Gasteil were supplied from outside with ready-to-cook and processed grain, either from adjacent communities or from a larger distance. This consumer character is in accordance with observation from previously analysed metallurgical sites. Interestingly, the components observed in charred cereal products (barley, *Hordeum vulgare*, and foxtail millet, *Setaria italica*) contrast with the dominant crop taxa (broomcorn millet, *Panicum miliaceum*, foxtail millet, and lentil, *Lens culinaris*). Foraging of fruits and nuts also significantly contributed to the daily diet.

plant remains are stored in the archaeological depot of the State Collections of Lower Austria and are available for scientific re-evaluation on request: Landessammlungen Niederösterreich, Bereich Urgeschichte und Historische Archäologie, MAMUZ/Schloss Asparn/Zaya, Schlossplatz 1, 2151 Asparna. d. Zaya, Österreich/Austria. E-mail: franz.pieler@noel.gv.at, phone: +43 (27 42) 90 05 – 499 12.

**Funding:** TJ, SW, and AGH received funding from the Austrian Science Fund (FWF) project "Life and Work at the Bronze Age Mine of Prigglitz" (proj. No. P30289-G25, PI Peter Trebsche) for the analysis of plant macroremains and charcoal. https://pf.fwf.ac.at/en/research-in-practice/project-finder/41105 AGH received funding from the Federal Government of Lower Austria for the analysis of plant macroremains and charcoal during the pilot project. http://www.noe.gv.at/noe/Kontakt-Landesverwaltung/Abteilung_Kunst-Kultur.html AGH received funding from the European Research Council (ERC) project "PlantCult" (ERC-CoG-2015, GA 682529, PI Soultana Maria Valamoti) for the SEM analysis of some of the cereal products. https://cordis.europa.eu/project/id/682529 The funders had no role in study design, data collection and analysis, decision to publish, or preparation of the manuscript.

**Competing interests:** The authors have declared that no competing interests exist.

# 1. Introduction

## 1.1. Miners as specialist producers—And as consumers

Mining for copper ores in the eastern Alps is documented from periods as early as the Late Neolithic, mostly in the area of today's Austrian federal states of Tyrol and Salzburg [1]. From there, copper mining activities spread rather slowly eastwards until they reached the very eastern Alpine foothills. The first archaeological traces of copper mining in these remote eastern Alpine mountain ranges have been dated to the Late Bronze Age [2, 3], a period characterised by generally intensified settlement activities across the Alpine range, which is commonly associated with the increasingly warmer climate after the "Löbben" [4] climate deterioration [5–7].

It is commonly accepted that, just as in later societies, the craftspeople involved in the *chaînes opératoires* (operational sequences) of Bronze Age copper ore mining, copper smelting, and bronze production—miners, smelters, foundrymen—were highly specialized [8]. Full-time specialization in mining, however, could only emerge within or supported by a community that provided them with everyday necessities, i. e. primarily food and raw materials. This hypothesis presupposes that a surplus of agricultural goods was produced in the surroundings of the mining and smelting sites and delivered to the working areas.

While a mining site is clearly a *producer site* for copper and bronze products, the aforementioned considerations put it into the position of a *consumer site* when it comes to food supplies. In the case of the Bronze Age salt mine of Hallstatt, the agricultural hinterland is located a several day's journey away in the river valleys or at the high alpine pastures of the Dachstein mountains [9, 10]. For other mining regions in the Alps, it has been suggested that supplies from both a local production and the hinterland may have been required for the maintenance of their productivity [11]. Archaeobotanical and palynological studies from Bronze Age mining sites have already highlighted various aspects of such models [e. g. 9, 11, 12, 13–17].

## 1.2. Pork... and what else? Current state of research into Alpine Bronze Age miners' diets

The animal-based part of the alimentation of the specialists involved in mining and metallurgy has already been exhaustively studied based on animal bone assemblages from mining camps, smelting sites, and contemporary producer sites. Archaeozoological analyses have shown that the meat supply of mining sites throughout the Eastern Alps was mostly based on pork production. A high percentage of pig bones is characteristic throughout the Early Bronze Age until the Late Bronze Age. Based on sex determination, the distribution of slaughtering ages, the representation of skeletal elements, and estimates of the areas required for the corresponding livestock farming, the pigs are assumed not to have been raised in the mountainous terrains but in more suitable environments; any surplus of pork would have been delivered to/exchanged with the miners. The Hallstatt salt mines even provided evidence for large-scale on-site pork curing [10, 18]. The results from Prigglitz fit into this general Bronze Age pattern [6, 10, 19] concerning the miners' obvious preference for pork for meat consumption. Recent studies have revealed a change in dominant meat categories at the beginning of the Early Iron Age [20].

Current insights into the plant-based aspects of Bronze Age miners' food are far less conclusive: While ample archaeobotanical evidence exists for Iron Age mining sites such as Dürrnberg [21–23] or the Iron Age phases of Hallstatt [24, 25], archaeobotanical data on Bronze Age mining in the Eastern Alps is still somewhat anecdotic and mostly restricted to the early centuries of copper mining (Fig 1).

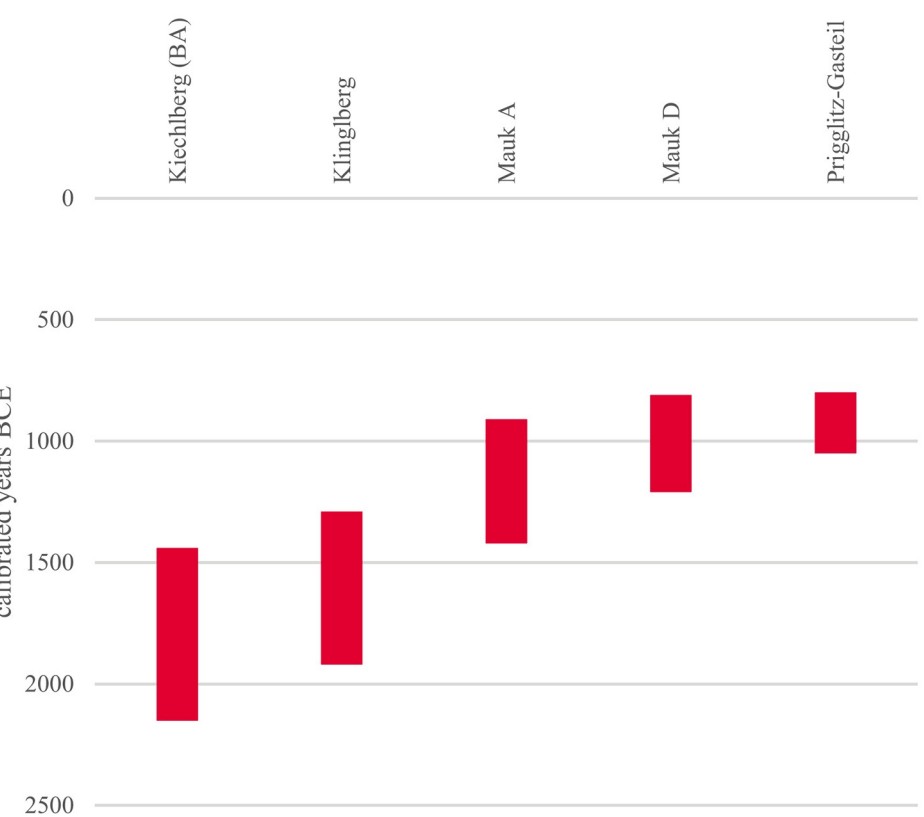

**Fig 1. Rough temporal ranges of the archaeobotanically investigated Bronze Age mining sites referred to in this paper.** The overview is based on the 2σ ranges of the oldest and youngest calibrated radiocarbon dates, respectively. Intra-site modelling was only considered for Prigglitz (see section 2.3.2). Published data from Kiechlberg [17], Klinglberg [26], and Mauken [27] were recalibrated with OxCal 3.10 [28] and the IntCal13 calibration curve [29]. Illustration: OeAW-OeAI/A. G. Heiss.

The multi-phase hilltop settlement at **Kiechlberg** (Thaur, Tyrol, Austria; western margin of mining region 5 in Fig 2) is outstanding due to its temporal range, as the oldest settlement activities relatable to copper metallurgy are as early as the Late Neolithic [30, 31]. A total amount of 20 flotation samples (c. 115 l) were taken for archaeobotanical analysis. While the samples retrieved from the Late Neolithic midden layers resulted in a rich account of more than 800 macroremains of cultivated crops [17], the Early to Middle Bronze Age layers which shall be considered as a comparison to our own study were mostly uninformative: The material contained only punctual evidence (n<5 each) of pea, hazelnut, and unidentifiable cereal grains [17], thus allowing no inferences on the organisation of food supplies. Finds of processed food-stuffs are not documented for any of the phases at Kiechlberg (Oeggl, pers. comm. 2020).

The Early Bronze Age hillfort settlement at **Klinglberg** (St. Veit im Pongau, Salzburg, Austria; mining region 8 in Fig 2) was not only the first site of this kind which was investigated archaeobotanically in the eastern Alps. With 1,803 soil samples and a total volume of 14,600 litres of processed soil, Klinglberg also represents the most intensively sampled and analysed Bronze Age copper production site from an archaeobotanical perspective, and the conclusions drawn have fundamentally influenced the current ways of how to look at food supplies for Bronze Age mining communities. Analysis of the charred plant remains resulted in a clearly emmer- and barley-dominated cereal spectrum, complemented by pea on the legume side [16]. The botanical find assemblage was characterised by an overall lack of cereal chaff in

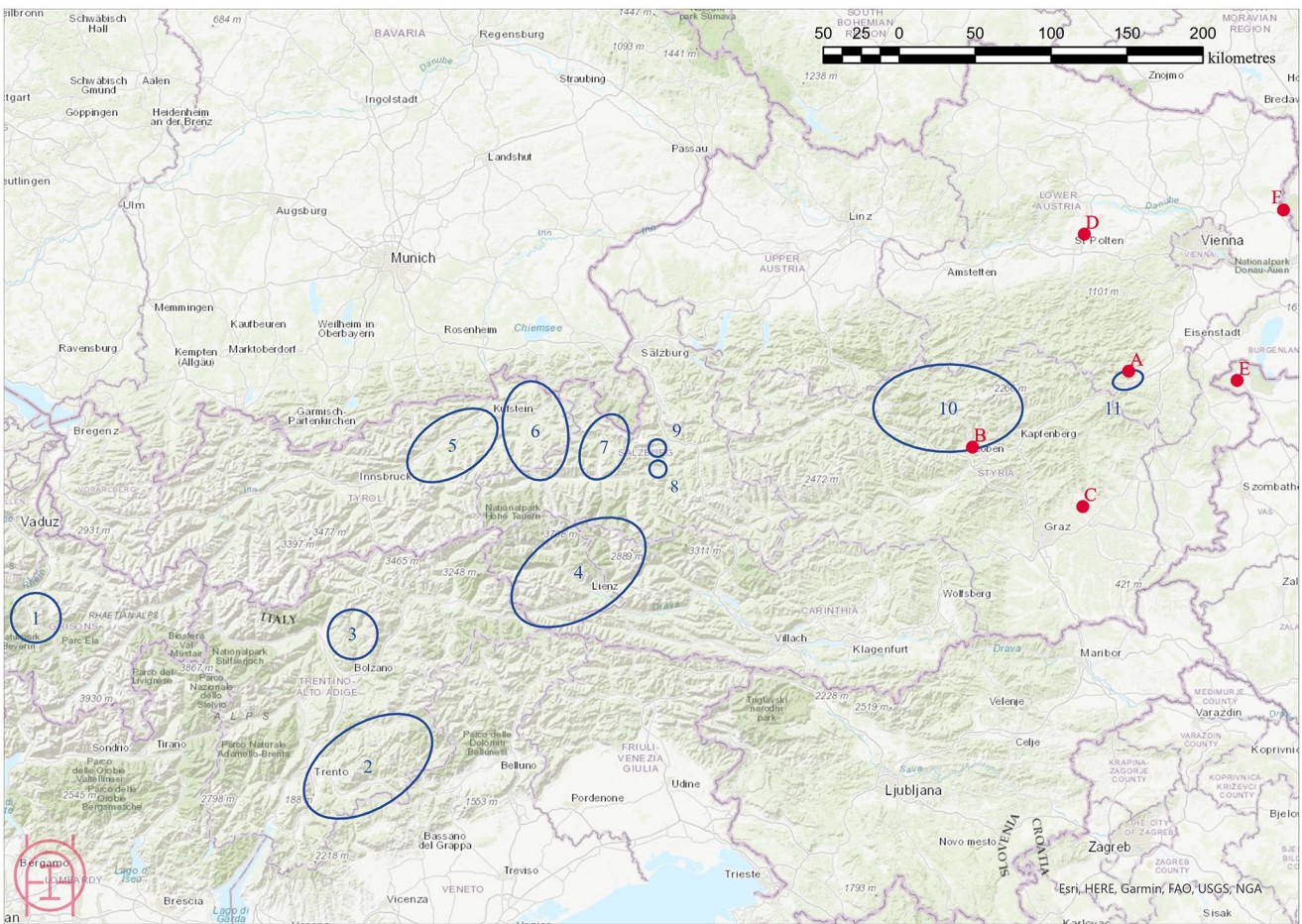

**Fig 2. Map of regions and sites mentioned in the text.** Blue circles: copper mining areas in the eastern Alps during the Bronze Age [2]: 1...Grisons, 2...Trentino, 3...South Tyrol, 4...East Tyrol, 5...Schwaz-Brixlegg, 6...Kitzbühel-Kelchalm-Jochberg, 7...Saalfelden Basin, 8...St. Veit-Klinglberg, 9...Mitterberg, 10...Upper Styria, 11...Prein-Prigglitz-Kulmberg. Red letters: Closest settlements with available archaeobotanical data contemporary to Prigglitz-Gasteil: A Prigglitz-Gasteil, B... Kulm/Trofaiach, C... Neudorf/St. Ruprecht a. d. Raab, D... Unterradlberg, E... Sopron-Krautacker, F...Stillfried a. d. March. Map: OeAW-OeAI/A. G. Heiss.

contrast to ample evidence of grain; furthermore, chunks of processed cereals were found in some of the sampled contexts, which were interpreted as chunks of charred bread by F. J. Green and S. J. Shennan. They considered the archaeobotanical find assemblage as indicative for grain supplies from outside the mining area, delivered in the state of bread and of "ready-to-cook" grain [16, 26, 32]. In-depth analyses of the "bread" remains have, however, not been carried out.

Prior to the current study, the mining and ore-processing site of **Mauken** (Radfeld, Tyrol, Austria; mining region 5 in Fig 2) has been the only Late Bronze Age mining site in the Eastern Alps providing plant macroremains. At the copper ore processing and smelting site Mauk A (Late Bronze Age, late 12th to 11th century BCE), slope water had created waterlogged conditions with excellent preservation of wooden implements. Still, the only cultivated plant remains found were a single charred grain each of broomcorn millet, hulled barley, and an imported possible condiment [6, 12, 33]. No traces of processed foodstuffs were found.

In the current paper, we present archaeobotanical data resulting from a high-resolution sampling approach applied during the fieldwork at the Late Bronze Age copper production

site of Prigglitz-Gasteil in the years 2010 to 2014. By comparing the results obtained from our own analyses with previously published data, we aim at improving the overall knowledge on plant-based food resources of Bronze Age miners in the Alpine range.

## 1.3. The copper production site of Prigglitz-Gasteil

**1.3.1. Location.** The Late Bronze Age site of Prigglitz-Gasteil (47˚42'46"N 15˚56'29"E) is situated in the modern cadastral area of Prigglitz, district of Neunkirchen, in the Southeast of what is today the State of Lower Austria. The area is part of the easternmost copper mining region known in the Eastern Alps (region 11 in Fig 2). The nearest Late Bronze Age settlements from which archaeobotanical data are available for comparison are more than 50 kilometres away (Fig 2): To the west, Kulm/Trofaiach [34], to the south Neudorf/St. Ruprecht a. d. Raab [35], to the north Unterradlberg [36], and to the east Sopron-Krautacker [37].

**1.3.2. Excavation and chronology.** The site of Prigglitz-Gasteil was discovered in the 1950s by F. Hampl [38], and it was soon recognized as the largest prehistoric copper mining site in Lower Austria. In 2010, author P. Trebsche resumed modern fieldwork resulting in five seasons of excavation (2010–2014) and several campaigns of geophysical prospections and core drillings (2017–2018).

During these excavation campaigns, an absolute chronology for mining activities has been established, placing them into the 11[th]–9[th] centuries BCE of the late Urnfield culture [3, 39]. With the aid of geoelectric and seismic measurements, complemented with core drillings, it was possible to reconstruct the Bronze Age mine working as a large opencast mine reaching a depth of at least 30 m below the actual surface. During the excavations, two terrain terraces immediately next to the opencast mine were investigated. The Bronze Age terraces had been cut into older layers of mining debris to create horizontal space for buildings and workshops.

Buildings were exclusively made of timber using different construction techniques, with wattle and daub walls. Many of the buildings contained hearths which could have been used either for domestic purposes or for metallurgical production. Numerous finds from the accumulated cultural layers indicate the refinement of copper, casting of bronze objects, bone and antler working as well as cooking and food consumption. The area investigated on terraces 3 and 4 (Fig 3) is therefore interpreted as the habitation of the mining community working there and/or the workshops of people supplying the miners [3, 19, 39–41].

**1.3.3. Geology, soils, and vegetation cover.** The excavation area is situated on the eastern slopes of the Gahns, a plateau connected to the Schneeberg massive, in the transition of the Greywacke Zone and the Northern Calcareous Alps (Styrian / Lower Austrian Limestone Alps). At the interface of these two geological units, copper ore (chalcopyrite) and iron ore (siderite) form the deposit of Gasteil "Sandriegel" [42]. The topsoils on the slopes in the research area are predominantly shallow ranker soils [43].

Vegetation historical data are not available for our area of interest up to now. Preliminary palynological studies of local vegetation history are however currently being undertaken, and an exhaustive study will be carried out during a consecutive project (see Conclusions and outlook). The available vegetation historical framework is therefore limited to observations of current vegetation, supported by models of potential natural vegetation [PNV, 44]. Current vegetation forms a mosaic of extensively cultivated fields as well as pastures and densely forested areas. The latter, as a characteristic for the transition between the influences of sub-Illyrian and Pannonian climates, are composed of spruce-fir-beech forests (Abieti-Fagetum) and black pine forests (Seslerio-Pinetum nigrae) [45, 46]. Occurrence of these two main forest types is congruent with the local potential natural vegetation [47].

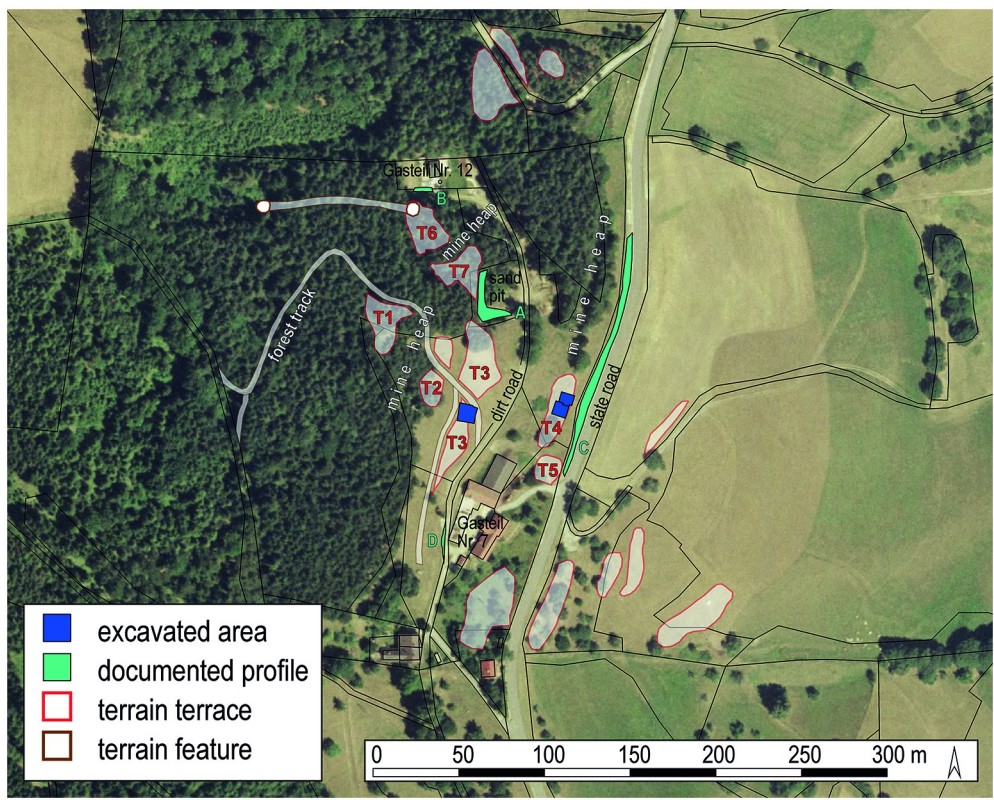

**Fig 3. Aerial photograph of the excavation site.** The investigated working platforms/terraces are indicated. Image: UIBK/P. Trebsche.

## 1.4. Research goals

The project "Life and Work at the Bronze Age Mine of Prigglitz", directed by P. Trebsche and running from October 2017 until September 2021, is currently investigating the operational sequences and flows of goods not only of the mining, smelting, and alloying products, but also those concerning tools, construction materials, fuel, and provisions with food. In the current study, the authors aimed at addressing the following aspects of the project:

1. To provide a theoretical framework for the evaluation of plant-based culinary artefacts (predominantly when in charred state) found in archaeobotanical find assemblages, basing on previous work by the first author [48].

2. To present the identified charred remains of food plants which were retrieved from the Late Bronze Age mining site at Prigglitz-Gasteil in a high-resolution sampling approach. The fragments of processed cereal preparations will be presented as the "missing link" between crop spectra and actual diet.

3. To evaluate the results from Prigglitz-Gasteil as compared to archaeobotanical data from four other Bronze Age copper production in the Eastern Alps, and from (supra-)regional crop spectra covering various types of settlements in the surrounding regions (eastern and southern Austria, and western Hungary), thereby exploring possible cultural, spatial, and chronological differences in nutrition and food processing patterns.

4. To reconstruct the *chaînes opératoires* of plant-based dishes found at Prigglitz-Gasteil, basing on the current bioarchaeological evidence.

5. To add up to the current state of research on food supplies for Bronze Age copper production sites in the Alps, with a focus on subsistence patterns and culinary aspects at Prigglitz-Gasteil.

## 2. Materials and methods

### 2.1. General statements

**2.1.1. Availability of data and material.** All relevant data are within the manuscript and its Electronic Supplementary Materials (ESM). All archaeological plant remains are stored in the archaeological depot of the State Collections of Lower Austria and are available for scientific re-evaluation on request: Landessammlungen Niederösterreich, Bereich Urgeschichte und Historische Archäologie, MAMUZ Schloss Asparn/Zaya, Schlossplatz 1, 2151 Asparn a. d. Zaya, Österreich/Austria. E-mail: franz.pieler@noel.gv.at, phone: +43 (27 42) 90 05–499 12.

**2.1.2. Ethics statement.** The individual in Fig 7, our valued colleague Michael Konrad, has given written informed consent (as outlined in PLOS consent form) to be depicted in the publication. No additional permits were required for the described study, which complied with all relevant regulations.

### 2.2. Approaching the miners' plant-based nutrition

*Off-site* data obtained from pollen profiles, where available, serve as a palaeoecological framework, providing diachronic information on vegetation history, agricultural activities, and sometimes mining-associated pollution [11, 14, 15, 49–52]. When approaching the agricultural foundations of a mining community, they are the most important tool in the reconstruction of land use patterns and past agricultural landscapes.

To gain insights on the alimentation itself, however, analysis and careful interpretation of *on-site* archaeological plant macroremains still play the key role, in particular when it comes to the characterisation of the producer or consumer character of a site [53–56]. Approaches towards such differentiation have been successfully carried out for numerous other contexts than mining, resulting in a variety of models and inferences [e. g. 54, 56–59] and, although rarely explicitly stated as such, all these models root in theoretical concepts on crop and food processing. Fig 4 illustrates such a general framework using a behaviour chain [60]–a concept that is more generalist and abstract than the more refined *chaîne opératoire*, and which is therefore more suitable for outlining general theoretical statements [61].

Building onto such simple principles, the pioneering works by G. Hillman [57, 67] and G. E. M. Jones [68] initiated the application of complex operational sequences obtained from ethnography and experimental observations onto archaeobotanical finds. The results enabled the modelling of detailed *chaînes opératoires* of crop processing basing on a seed assemblage's composition of grain, chaff, and weed seeds, respectively (Fig 5) [54, 56, 69–71].

However, operational sequences modelled for edible plants usually end with the stage of ready-to-cook/ready-to-eat grain or seed. Reconstructions of the "biographies of things" [74] in archaeobotany usually leave a huge blank space between a crop and its consumption. This space leaves nothing less unexplored than the huge field of *cuisine*, the "cultural domain which is principally concerned with the knowledge and behaviour of a given cultural community regarding the preparation and consumption of food" [75]. In the case of the term "cooking", we follow S. Graff as contrasted to e. g. K. E. Twiss [76] in her preference for a clear terminological differentiation between *cuisine* and "cooking", thus limiting the latter to the "food

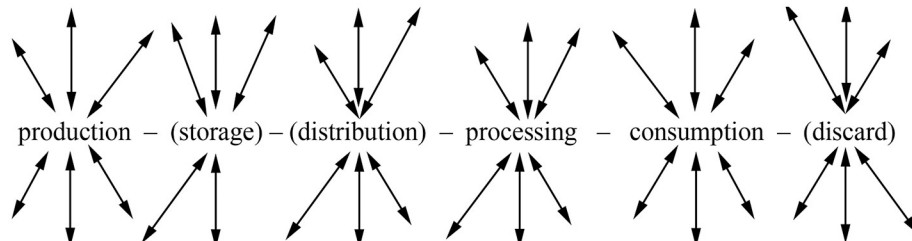

**Fig 4. Human-thing interactions along a generalised behaviour chain for food items, basing on a concept brought forward by I. Hodder [62, 63] and adapted to considerations specific to human-food interactions [64–66].** Activities in brackets are facultative and mobile elements, which can take place in virtually any position of the chain. Illustration from Heiss [48].

**Fig 5. Illustration of a *chaîne opératoire* for crop processing using the example of hulled wheats.** The sequence starts on top with the cereal plant, proceeding counter-clockwise towards "ready-to-cook" grains. Ramifications (optional steps) in dashed lines. Illustration basing on the original design by Stevens [54], additional steps such as harvesting, kiln-drying, storing, and parching/soaking prior to dehusking were added [65, 72, 73]. Image from Heiss [48].

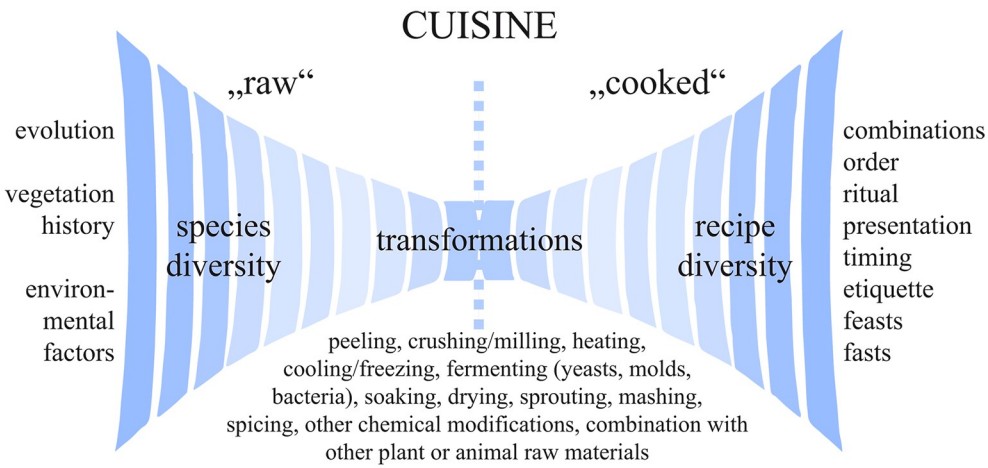

**Fig 6. *Cuisine*, the art of cooking, illustrated as a space of possibilities.** The transformations of natural, unchanged ("raw") plant components into culturally modified ("cooked") food is roughly inspired by Lévi-Strauss' [78] *triangle culinaire* (*cru–cuit–pourri*). While natural conditions (on the left side) influence the possible ingredients, it is cultural aspects (on the right side) on which their processing and consumption patterns depend. Image: OeAW-OeAI/A. G. Heiss, modified from Katz and Voigt [79].

preparation strategy that involves the application of heat (. . .) such as boiling, roasting, baking, frying, or smoking" [77]. In Fig 6, we set the term "cooked" under quotation marks for this reason.

*Cuisine* as the "elephant in the room" has however mostly not been ignored by archaeobotany out of negligence, but due to serious methodological constraints: As soon as a grain or seed loses its shape when crushed or ground, not only is it transformed a big leap further towards becoming an artefact, but the resulting archaeological remains become much more difficult to identify, and to interpret [48, 80, 81]. On the upside, the resulting materials bear potentially legible traces of the processes they have undergone [48, 82, 83]. The following section outlines a few important aspects which, to the authors, seem to be of importance when approaching these difficulties.

**2.2.1. Looking at food remains as culinary artefacts.** While the role and relevance of plant remains as parts of a certain material culture are undoubtedly commonly accepted [84, 85], it is particularly the dichotomy between ecofacts and artefacts which might still require a closer look when dealing with processed food remains, as "ecofacts typically are not considered artefacts unless clear indications exist that they were modified" [86]. Finds of harvested grain can therefore certainly be regarded as ecofacts. However, what if grain is ground into flour, mixed with water, shaped into a bread, and then baked? We are convinced that such an object is clearly artefactual [48]: Archaeological finds of processed foodstuffs are the remnants of objects "predominantly shaped by human action" [87], and they precisely match common definitions of what is an artefact [88–91]. They are the material outcomes of human action and creativity, they are biogenic artefacts produced within the boundaries of a certain *cuisine* [75]. They are the results of transformations, which replace natural shapes and compositions by culturally determined ones (Fig 6). They bear not only information on the raw materials they contain, but also on the processes these materials have undergone. We therefore entirely disagree with approaches regarding remains of processed foodstuffs as non-artefactual [e. g. 87].

Before proceeding, it therefore seems necessary to briefly continue arguing against definitions of artefacts and their delimitations to ecofacts that acknowledge a piece of chipped flint as the artefactual outcome of a skilled maker's action [92, 93], but not a bread bun or a jug of

beer (the beer, not the jug), which both result from highly complex production processes [94–99]. In the following, we point out a few possible explanations for such—in our opinion strangely distorted—views:

1. **Preservation biases**. Renfrew and Bahn [100], as an example, seem to confirm this by the contextual association of "non-artefactual" with "organic" and "environmental remains" as a matter of course. However, despite its equally organic nature and the same resulting difficulties in preservation, wood seems to be entirely independent from such conceptual restrictions: wood is commonly regarded as a raw material perfectly suitable for things considered as artefacts [100–102]. Furthermore, food's intended destiny lies in consumption. It could therefore even be its ephemeral nature that, in the eyes of some, disqualifies food remains from being regarded as meaningful elements of a certain material culture.

2. **Methodological biases**. As mentioned in the previous section, the understanding of archaeological finds of processed food is intrinsically tied to the analyses of their inner structures [82, 103–107]. Original surfaces are very often missing, and even complete objects (which rarely occur) are usually neither as showy nor as straightforward to describe as it is the case with a pot, a brooch, or a glass bracelet. S. R. Graff [77] ironically referred to food remains as "boring artifact categories". Furthermore, stringent typologies for archaeological finds of foodstuffs are still widely not available [48, 82].

3. **Social and gender biases in research**. In her article, Graff also points out that presuppositions on the producers of food—women, maybe also children or slaves, as she summarises it—have rendered the outcomes of *cuisine* as something "viewed as less valuable by the scholarly community" [77], and have thus limited archaeological research interest (and endeavour) of the entire topic for a long time [see also 64, 108, 109].

Very likely, all three types of biases have, to varying extent, contributed to the "non-artefactual" scholarly look towards archaeological food, challenging of which has even been regarded as "blurring and subverting these boundaries [between artefact and ecofact]" [87] instead of being judged as consistent. As a concluding statement, we may quote from A. Sherratt's 1991 publication that "People don't eat species, they eat meals" [110] and add ". . . and these meals are culinary artefacts."

We believe that acknowledging the look at archaeological remains of processed food as the artefactual outcomes of a past *cuisine* is the prerequisite to accept not only the possibility but the necessity to describe the production of an archaeological dish in a specific *chaîne opératoire* [64], regardless (but aware) of the possible limitations. Just as for any other type of artefact, a better understanding of the transformative processes applied within a past *cuisine* will also allow setting foundations for the creation of typologies of archaeological foodstuffs not only basing on their mere outer shapes, but also encompassing their components and the operational sequences involved in their production [48].

However, any attempt to reconstruct both the ingredients and the *chaînes opératoires* of their processing—that is, the recipe—for a charred archaeological food remain must inevitably remain incomplete. The primary source of mischief lies within food preparation itself: Aside from their numerous effects on texture, taste, and shelf life, many food processing techniques directly aim at facilitating the consumption of raw food materials that would otherwise be difficult, if not impossible, to digest [111, 112]. From the researcher's view, the higher the degree of refinement, the lower the chance to identify the ingredients under a microscope, because fragmentation gets ever stronger, while diagnostic elements increasingly disappear [e. g. grain husks and seed coats, see 107].

Many of the actions involved in food processing—most notably crushing/grinding, cooking/boiling, and fermenting [113]–do not only pre-digest the raw ingredients for humans, but also for any other hungry life form. Their resistance to microbial attack is considerably reduced, with the consequence that flour and its products decompose rapidly, even under waterlogged conditions [98, 114].

**2.2.2. Prerequisites for preservation, and for analysis.** There are environmental conditions which allow for the preservation of chemically unmodified processed foodstuffs over archaeologically relevant periods of time, for example if desiccation [115–117] or high salt concentrations [118, 119] are inhibiting microbial growth. Outside such contexts, however, the analysis of archaeological cereal products is mainly limited to charred material [82, 120].

Charring is well-known for being a strong filter for plant remains not only due to the charring conditions themselves [121–123], but also because it limits the preservation of processed plant foods to particular events such as "baking accidents", intentional burning [106], and catastrophic fires [82, 83]. Furthermore, once charred, the material becomes brittle and highly sensitive to mechanical stress, the presence of which influences the chances of preservation—particularly of larger objects—during deposition and recovery [48, 120]. On the positive side, charring does not generally affect the determinability of plant tissues, as not only the cell wall structures [104, 105] but also subcellular elements such as starch granules can remain recognizable in charred state [124–126].

Charring massively transforms the chemical composition of organic matter [127–129] and consequently places narrow limits on chemical residue analyses. They are still possible, but in contrast to histological approaches they are often limited to very general statements when it comes to charred plant-based materials [107, 130–137]. Chemical analyses of uncharred (that is, desiccated) food preparations have, in contrast, already revealed their potential and diagnostic acuity [138, 139].

The rather recent field of (archaeo-)proteomic analyses, applied to (partially) charred remains, could spark a revolution in the near future: In archaeological contexts, proteins appear to be more resilient to decomposition than e.g. DNA [140, 141], and they have already proven to not only identify organisms but also their specific tissues with unsuspected precision [142]. As for the site of Prigglitz-Gasteil, lipid residue and proteomics analyses of suspected cooking ware sherds are envisaged for a follow-up project.

**2.2.3. Rough guidelines for the work with charred fragments of food preparations.** Roughly summing up the previous two sections and pages 39–50 from the first author's habilitation thesis [48], we think that the analysis of archaeological food preparations should always be accompanied by the following considerations:

1. **Limited visibility of plant tissues**. Non-destructive SEM analysis can only be carried out on surfaces (of subsamples), while any plant tissues enclosed within the charred foodstuff (e. g. a cereal product) remain invisible. Sometimes, even the expected main ingredients (e. g. flour) in a sample do not deliver enough identifiable material, while possible accessory components—accidental ones such as glumes, or intentional ones such as condiments—are even more difficult to track. However, there are cases where even condiments have successfully been identified [82, 143, 144].

2. **Limited identifiability of plant tissues**. Archaeobotanical identification bases on intact cell wall structures, and thus on robust thick-walled tissues. Consequently, outer hulls (e. g. chaff, seed coat, pericarp) often preserve. Thin-walled storage parenchyma (cereal endosperm, pulse cotyledonary tissue, fruit pulp) frequently collapses and fuses into amorphous masses when charred. The higher the degree of refinement, the lower the chance to find anything identifiable (see section 2.2.2).

3. **Visibility of many components only to chemical residue analysis**. Any ingredients not leaving distinct cell patterns remain invisible to histological approaches towards charred material. Solid and detectable animal parts such as fish scales [107] only rarely make it into processed foodstuffs, while meat, lard, and dairy products leave no identifiable traces other than chemical ones. However, due to charring, chemical diagnosis can be very limited (see section 2.2.2).

4. **Different processes can lead to identical structures**, recently subsumed by J. J. García-Granero under the term of **equifinality** [145]. As an example, observation of intact starch granules in a charred cereal product is clearly indicative of its charring in dry state [48]. In contrast, the absence of intact starch granules is ambiguous and may either indicate charring in hydrated state, or charring of an already pre-cooked/pre-boiled product (which may or may not have been subsequently dried prior to charring) [103].

5. **Complete identification of components is currently impossible** (see above). Even easily identifiable plant-based components may be hidden in the material, and most animal-based components will go undetected without chemical residue analysis.

6. **Culinary production processes frequently involve recycling**, which is why every product can basically serve as raw material for another product. Bread, for instance, can be dried and stored as a food preserve, eventually becoming the basis of a soup or stew [146]. Ground dry bread can be mixed into fresh bread dough [147, 148], or hydrated for the production of *kvass*-like beers low in alcohol [149–151]. After mashing, the spent grain can in turn be used as a starter for making bread [66, 152].

All in all, **quantitative conclusions on ingredients should not be made**, and it is important to consider that **composite foodstuffs may not be recognised as such**. The possibility of **complex, even iterating operational sequences** must always be kept in mind. Against this background, any modelled *chaîne opératoire* should be accompanied by an **open discussion of potential flaws and uncertainties** [48], in order to avoid misinterpretations.

## 2.3. The material from Prigglitz-Gasteil

**2.3.1. Sampling and sample processing.** During the excavations from 2010 to 2014, all stratigraphic units (*Stratigraphische Einheiten*, SE) were systematically sampled. The archaeobotanical samples for the current study were taken from the residential area of the miners and/or craftsmen located immediately next to the mine on the two artificial terraces T3 and T4. From all occupation layers or building horizons, samples with volumes ranging from 10 to 20 l per sample were taken. Up to 30 samples per layer were taken from cultural layers that extended over larger areas, using a 1 x 1 m grid (Fig 7). The mining debris deposited between occupation phases was generally not sampled due to its almost complete lack in archaeological finds and visible plant remains (e. g. charcoal fragments).

Following this strategy, altogether 310 sediment samples with a total volume of 4,793.5 litres were taken from the excavated areas on terraces T3 (102 m$^2$) and T4 (113.45 m$^2$). This high-resolution sampling strategy was chosen to allow for spatial (Fig 7) as well as temporal (Fig 8) investigation of activities. All sediment samples were flotated and subsequently wet-sieved to retrieve botanical macroremains and micro-refuse such as casting droplets and bone fragments. Flotation was carried out with the flotation device set up at MAMUZ Schloss Asparn/Zaya in Lower Austria according to standard methodology [153, 154]. Sieve sets with mesh sizes of 2.0, 1.0, and 0.5 mm were used.

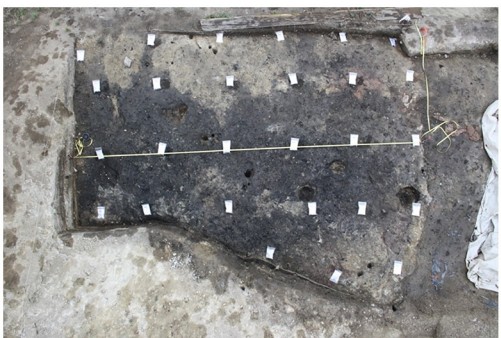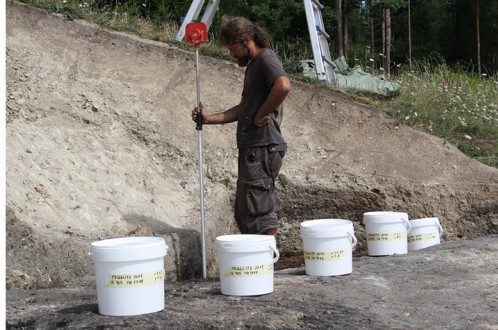

**Fig 7. Systematic sampling for botanical remains and micro-refuse by M. Konrad.** Images: UIBK/P. Trebsche.

**2.3.2. Dating and chronology.** The absolute chronology of the site is based on a large series of more than 75 radiocarbon dates funded within the scope of the FWF project, and mostly coming from short-lived organic materials (charred plant remains and animal bones), which were then calibrated in a stratigraphic model.

The habitation remains on Terraces 3 and 4 from which the archaeobotanical finds presented in this paper originate can be dated to the Late Bronze Age (c. 1300–800 BCE), more precisely to the Late Urnfield Period (c. 1050–800 BCE). Building activities on Terrace 3 and 4 overlap, but cover different time spans, with the activities on Terrace 3 starting in the second half of the 11th century BCE and ending at the beginning of the 8th century BCE, while the constructions on Terrace 4 only cover the last quarter of the 10th century BCE.

The upper Terrace T3 yielded a stratigraphy comprising eleven phases from the Late Bronze Age (phases T3-11 to T3-05), followed by a phase of erosion (T3-04), one phase of medieval

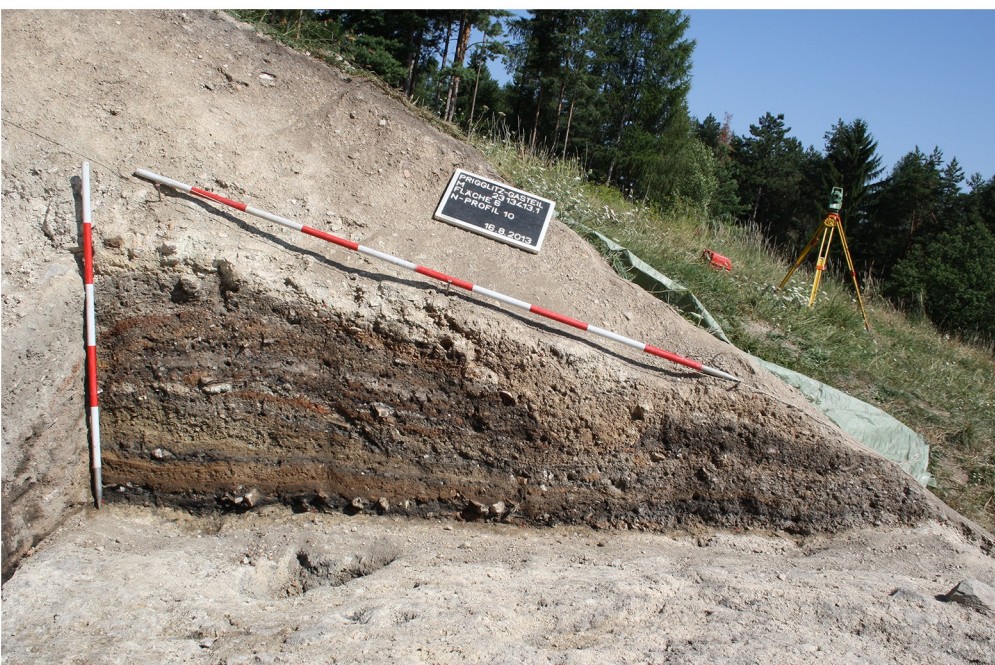

**Fig 8. Photograph of the sediment layers on Terrace 3 (T3) at Prigglitz-Gasteil.** Excavation area 6, profile 10. Image: UIBK/P. Trebsche.

occupation (phase T3-03) and two phases of the Modern Period (phases T3-02 and T3-01). According to a series of 19 radiocarbon dates, prehistoric activities in this area started from 1072–999 BCE (1σ) and lasted until 796–764 BCE (1σ), i. e., approximately two and a half centuries [3, 155].

The stratigraphy excavated on the lower Terrace T4 comprises 14 phases from the Late Bronze Age (phases T4-14 to T4-09) to the Medieval Period (phase T4-04) and the Modern Period (phases T4-02 and T4-01). During the Late Bronze Age, ten consecutive construction activities are attested at this terrace; they were interrupted by three episodes of copper ore mining [155]. Bayesian modelling of 13 radiocarbon dates [156] for short-lived organic materials allowed for a precise dating of the Late Bronze Age phases. The boundary start ranges from 946–906 BCE (1σ) with a peak in the probability distribution at 920 BCE. The boundary end date ranges from 910–851 BCE (1σ) with its peak at 895 BCE. Most probably, the entire stratigraphic sequence from the first construction in sub-phase T4-13G to the latest cultural layer deposited in phase T4-08 was created in only 25 calendar years [3, 155].

In addition to the high-resolution radiocarbon dating approach within our project, $^{14}$C dating of a single broomcorn millet (*Panicum miliaceum*) grain of from find no. 691 (SE 413), funded by the German Research Foundation (DFG) SFB/CRC 1266, was included into a recent overview of the early chronology of millet cultivation in Europe [157].

Basing on all available chronological information, a representative subsample of 90 soil samples from 74 stratigraphic units (SEs) with a total volume of 1,459 litres was selected from all available samples to get a diachronic overview of the Late Bronze Age activities on terraces T3 and T4. The samples represent nearly all phases of the Late Bronze Age occupation at the Prigglitz-Gasteil site (Fig 9).

**2.3.3. Laboratory work.** Heavy fractions from flotation and wet-sieving samples were searched for micro-refuse under a magnifier lamp, any charred plant remains which had remained therein were added to the light fractions. From all latter (organic) fractions, charred plant macroremains were sorted under the stereomicroscope (Olympus SZX10, magnification 6.3–63x), with the exceptions of wood charcoal fragments and entirely unidentifiable amorphous charred objects [82, ACOs, cf. 158, equivalent to the term 'AOV', cf. 159]. These two categories were only retrieved and counted if their grain sizes were 2 mm or larger.

During sorting, ACOs which displayed possible traces of plant tissue were consecutively checked for identifiable cereal tissue under an Olympus BX53M metallurgical microscope (magnifications of 100x up to 500x) in order to support their interpretation as cereal products [82]. Ten random fragments (highlighted in red in S1 Table) containing cereal tissue visible under the light microscope were selected for SEM analysis in order to investigate their components and inner structures.

SEM imagery for three of the ACOs (from finds no. 5, 8, 42) was produced at the Department of Molecular Botany of the University of Hohenheim within the scope of the ERC project PlantCult, using a Zeiss DSM 940 SEM after sputter coating the samples with gold/palladium in a Balzers SCD 040. Seven more ACOs (from finds no. 945, 1008, 2148, 2150, and 2153) were analysed at the archaeobotany laboratory of the Austrian Archaeological Institute (OeAW-OeAI) using a Hitachi TM4000Plus SEM without prior sputter coating. Species/genus identification of cereals followed common microstructural features of glumes and bran [160–162] as adapted to archaeological plant remains [83, 104, 105, 163, 164].

Identification of all other botanical macroremains, except for charcoal (see Conclusions and outlook) was carried out basing on the reference collection at OeAW-OeAI [165], general literature for seed identification [166–170], and specialised literature focusing on morphological cereal identification [171, 172].

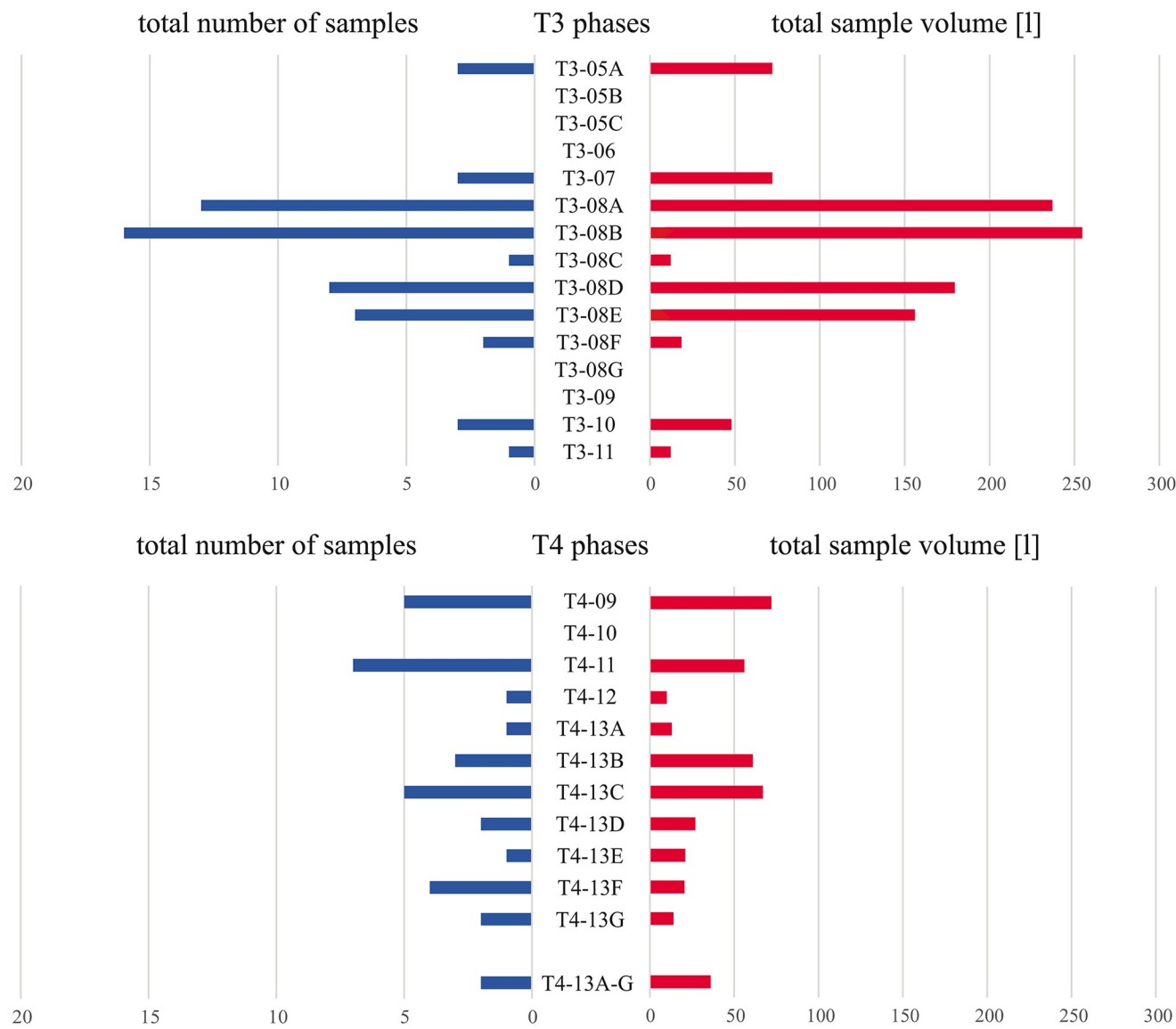

**Fig 9. Representation of the Late Bronze Age phases of a) terrace T3 and b) terrace T4 in the sample material.** For detailed values, please refer to the raw data in S1 Table. Image: OeAW-OeAI/A. G. Heiss.

Entire seeds as well as their fragments were all counted as one find each (see S1 Table). The only exception were the large quantities of fragmented charred conifer needles. To avoid over-representation, only the minimal numbers of needles were counted (see Table 1). Original counts of all fragments are, however, available on request.

**Table 1. Counting method applied to conifer needles.** Table from Heiss [173], modified.

| Number observed | Number counted |
| --- | --- |
| 10 entire needles | 10 |
| 10 tips | 10 |
| 10 bases | 10 |
| 10 bases and 5 tips | 10 |
| 10 tips and 50 middle parts | 10 |

**2.3.4. Data evaluation and documentation.** The results were recorded and evaluated using the ArboDat 2016 database [174, 175]. Phytosociological class groups [176], roughly represented in ArboDat as "ecogroups", were modified by aspects of the site's current vegetation observed during a vegetation survey in September 2010, carried out by M. Kohler-Schneider and A. G. Heiss. The resulting groups served as a basic means for classification of the habitats from which the identified plant remains could have originated. The occurrence of identified plants was evaluated by sample and by phase in this publication, basing on the respective total sum and ubiquity [frequency of occurrence, cf. 121].

Comparative diagrams of cereal spectra follow the guidelines proposed by Stika & Heiss [177, 178], i. e. grain finds of taxa unequivocally or at least probably (cf.) identified to species level are included, while identifications to genus level and above are excluded. Naked wheats which are not satisfactorily discernible by their grains are treated as a single species. Chaff finds are generally excluded from the diagrams. The resulting percentages are rounded to whole numbers. Ternary diagrams were created using the software Triplot [179].

Light micrographs were created using an integrated Olympus system (stereomicroscope SZX10, digital camera UC909, software Stream Basic), processed in Adobe Photoshop CC, and mounted into plates in Adobe Illustrator CC. The map in Fig 2 was created using ArcGIS Pro [180] basing on the following map sources: Esri, HERE, Garmin, FAO, USGS, NGA.

# 3. Results

The Late Bronze Age samples contained a total amount of 7,022 charred plant macroremains, retrieved from a soil volume of 1,459 litres. For the raw data, please refer to S1 Table. The mean find density of all plant macroremains (charcoal excluded) which amounts to 4.81 finds per litre (median: 2.3) is not particularly high. Nearly half of the botanical macroremains (n = 3,392) were not identifiable due to insufficiently preserved morphologies and/or surface features (Fig 10). Possible reasons leading to this extremely high proportion will be discussed later. The identifiable archaeobotanical finds (Fig 11) are mainly represented by seeds/fruits of cultivated plants (two thirds) and conifer needles (one third). Arable weeds as well as plants from woodland margins occur to minor extents, amounting to 10–12% each.

## 3.1. Cultivated crops

**3.1.1. Seeds and grains.** Domesticates, mainly cereals, occurred quite numerous (n = 1,116) and highly ubiquitous (71%) in the material, yet very unevenly distributed: 75 samples out of 90 contained less than one remain of cultivated plants per litre or even none at all. In contrast, there are three samples around, or well over, ten finds per litre: Sample 1421 (SE 820, phase T3-08B) with 9.9 finds/l, sample 2148 (SE 1047, phase T4-13E) with 21 finds/l, and sample 2153 (SE 1068, phase T4-13F) amounting to 18 finds/l.

Looking at the cereals only, 98% of the finds were small fragments with abraded surfaces, and they could not be assigned to any genus. The identifiable fraction of grain finds was clearly dominated by millet caryopses (Figs 12a and 12b and 13), with broomcorn millet (*Panicum miliaceum*) being more common at the site than foxtail millet (*Setaria italica*); whereas both taxa regularly occur in the samples taken at Terrace 3, they are nearly absent from Terrace 4.

Barley (*Hordeum vulgare*), emmer (*Triticum dicoccum*), and an unidentified wheat species (*Triticum* sp.), possibly poorly preserved emmer, occurred in minor amounts. Cereal remains other than grain (fragments) were rare and low in numbers: Culm fragments were the most common ones, deriving mostly from a single sample from terrace T4 (stratigraphic unit SE 1047), followed by a few glume bases of einkorn (*Triticum monococcum*) and emmer (*Triticum dicoccum*).

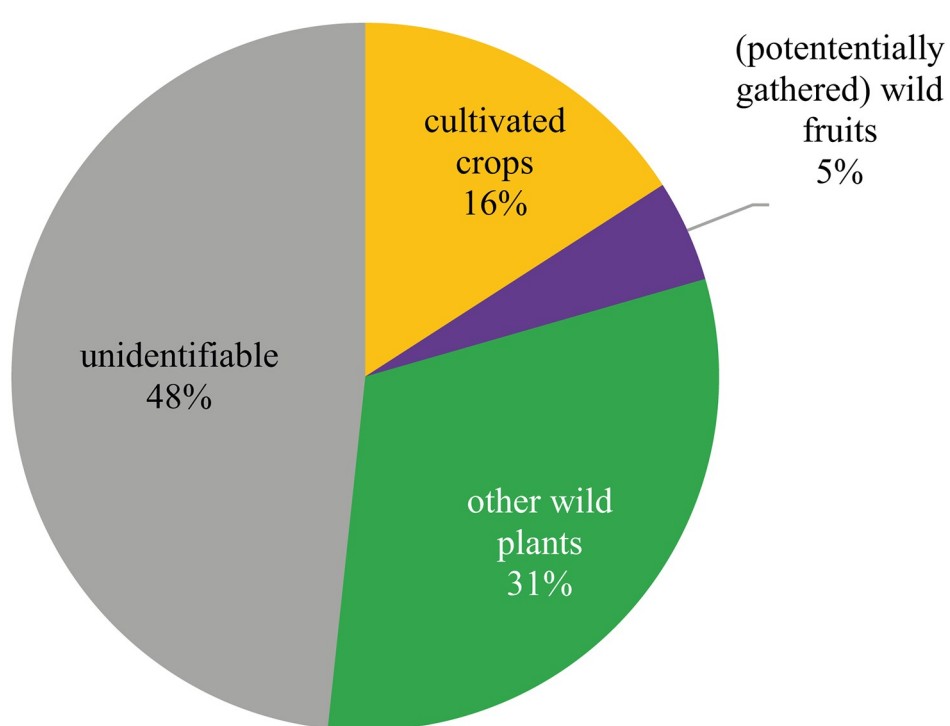

**Fig 10. Overall composition of the Late Bronze Age charred archaeobotanical find assemblage at Prigglitz-Gasteil.** n = 7,022. Diagram: OeAW-OeAI/T. Jakobitsch.

Finds of pulses were in general much rarer (n = 19) and far less ubiquitous (8%) than cereals at Prigglitz-Gasteil, lentil (*Lens culinaris*) being the only identifiable taxon.

**3.1.2. Cereal products.** A major group of cereal remains (n = 76, ubiquity 28%) was represented by amorphous charred objects (ACOs) containing fragments of cereal bran or glumes embedded in their matrix, and which are commonly interpreted as cereal products [48, 82, 120, 143, 181] (Fig 14). It was taken care to verify that the cereal tissue fragments were indeed contents of the chosen ACOs and were not just sticking to them. This was important because the endosperms of both barley and broomcorn millet tend to liquefy under certain charring conditions [182, 183], and create an amorphous matrix around otherwise intact grains [see also the experiment in 173]. This was not the case in the analysed ACOs.

Among the components, only barley (*Hordeum vulgare*) as well as foxtail millet (*Setaria italica*) were identified by their characteristic tissue features (Table 2, Fig 15), while any other possible ingredients must currently remain unknown (see section 2.2.3). None of the investigated samples displayed traces of intact (= ungelatinised) starch granules. The material was mostly very dense with only a few areas showing pores, but none exceeding 200 μm, qualifying them as micropores [184]. Due to the overall small sizes of the cereal product fragments, it was not possible to assess the size classes of a sufficient number of grain chunks contained therein. No information on the degree of milling/grinding is therefore available. Cell wall thicknesses observed in the preserved aleurone tissue did not show any obvious anomalies, which is why the authors did not pursue the issue of possibly malted grain [cf. 83] any further.

Numerous amorphous charred objects (n = 1,160) were listed in the group of "Indeterminata" (see S1 Table), as they did not display any obvious fragments of cereals or other plant remains on their surfaces. While future in-depth analysis of their inner surfaces may very well

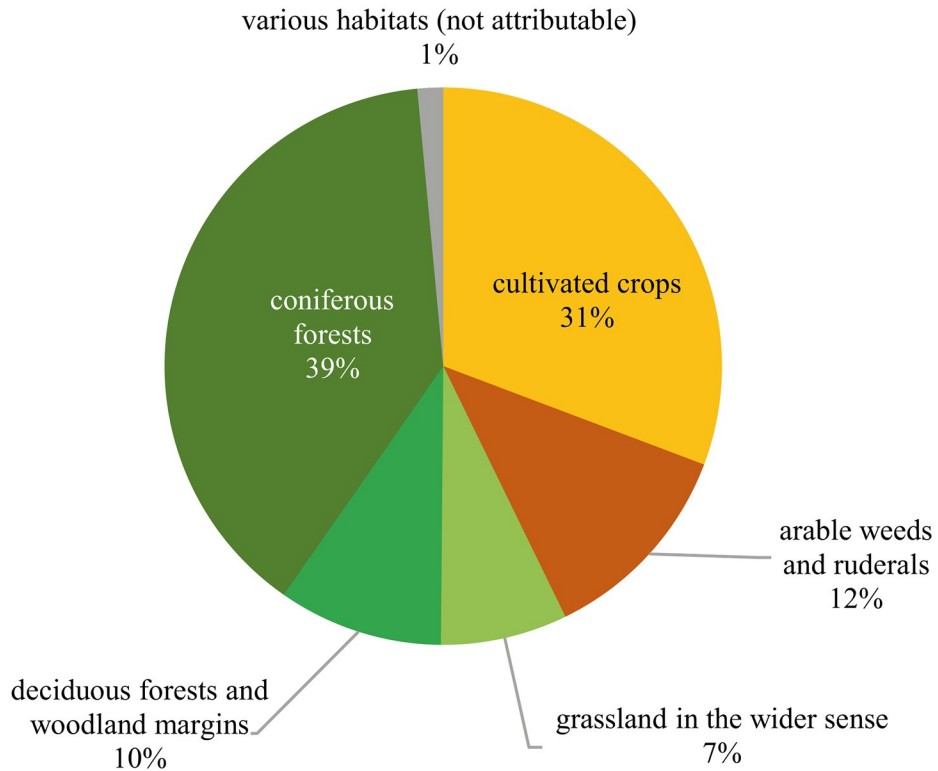

**Fig 11. Identifiable plant macroremains from Late Bronze Age Prigglitz-Gasteil, assigned to ecogroups.** The group "deciduous forests and woodland margins" overlaps to a large extent with the "wild fruits" group from Fig 10. The group "coniferous forests" is mainly represented by fir and spruce needles. n = 3,630. Diagram: OeAW-OeAI/T. Jakobitsch.

uncover more cereal products in this group, this must be regarded as speculative at the current state of research.

**3.1.3. Wild plants.** Among the remains of wild plants, the largest number of finds was attributable to the ecogroup of arable weeds and ruderals, i. e. taxa commonly restricted to anthropogenic habitats. This group (n = 437) occurred ubiquitous (64%) across the excavated areas of Prigglitz-Gasteil. The most frequent weed taxa belong to members of the goosefoot family (Chenopodiaceae = Amaranthaceae p. p.) and to wild millets (Poaceae-Panicoideae, including e. g. *Echinochloa* and wild *Setaria* taxa). These wild millets showed an overall distribution pattern similar to the one of cultivated millets.

Remains of species which were associated to open land ecosystems, and thus grassland in the widest sense (n = 267) were found to occur about as ubiquitous as weedy plants. The most numerous and ubiquitous group therein were Poaceae caryopses which were not identifiable any further, while the most common genera were bedstraw (*Galium* sp.), strawberry/cinquefoil (*Fragaria* sp. and *Potentilla* sp.), and sedges (*Carex* sp.). Plants from dry soils such as thyme (cf. *Thymus* sp.) were present, but also wetland taxa such as bugleweed (cf. *Lycopus europaeus*).

A variety of wild plants which preferably grow in light forests and on woodland margins was found (n = 329) (Fig 16), the most common remains being rose (*Rosa* sp.) nutlets and rosehip fragments (in total 154 finds), followed by drupes of the genus *Rubus* (n = 93). Other common finds from this ecogroup were hazel (*Corylus avellana*), crab apple or wild pear (*Malus* sp. / *Pyrus* sp.), sloe (*Prunus spinosa*), Cornelian cherry (*Cornus mas*), and elder (*Sambucus nigra* and *S. racemosa*).

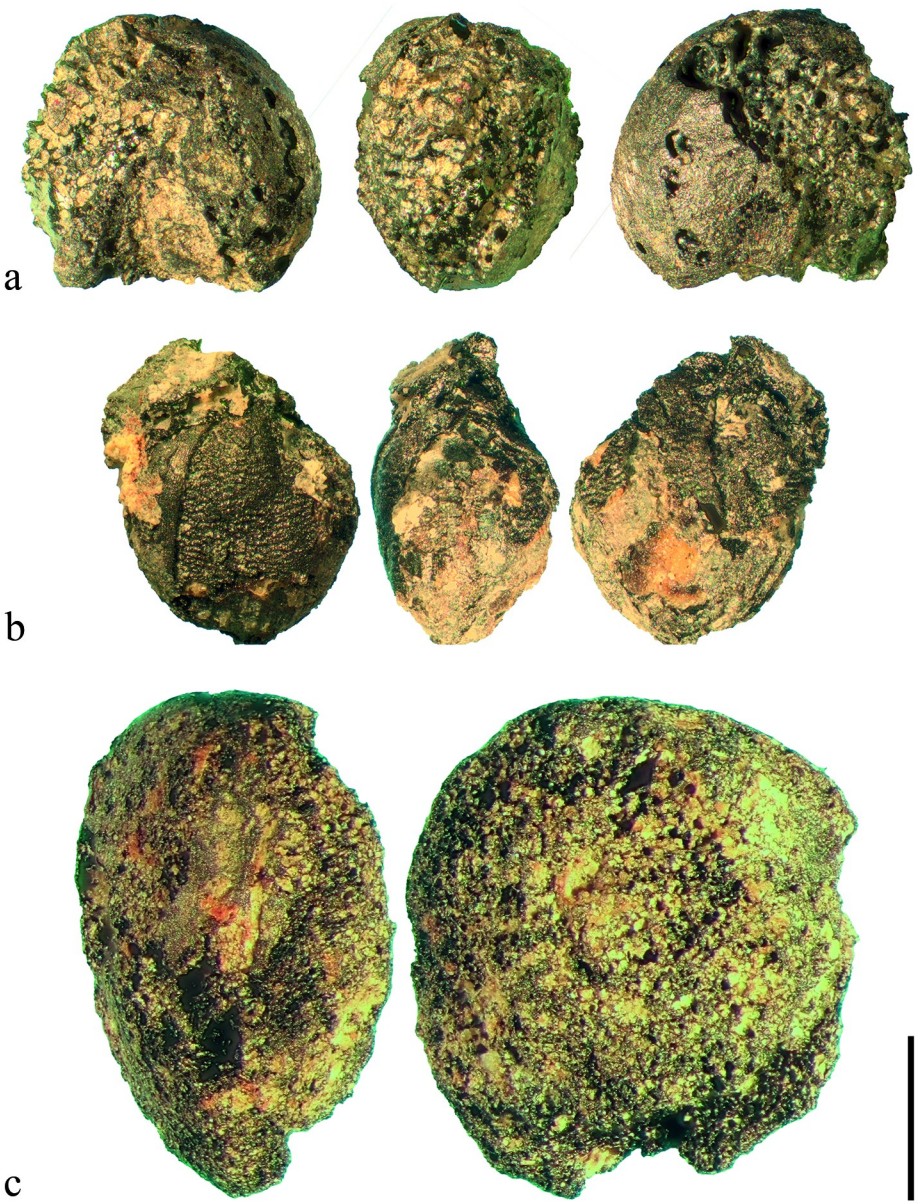

**Fig 12. Charred finds of the most important cultivated crops from the Late Bronze Age layers at Prigglitz-Gasteil.**
a) broomcorn millet (*Panicum miliaceum*), b) foxtail millet (*Setaria italica*), c) lentil (cf. *Lens culinaris*). Scale bar
length: 1 mm. Images: OeAW-OeAI/S. Wiesinger (top and middle row), A. G. Heiss (bottom row).

As mentioned above for the cereal products, it is likely that the large number of unidentified
ACOs also contains more of the parenchymatous fruit fragments which were sometimes
attributable to *Rosa* species and to *Malus*/*Pyrus* species. However, a complete overview of the
taxa contributing to the ACO will require a large-scale SEM approach.

Fir (*Abies alba*) needle fragments nearly exclusively represented the largest ecogroup in the
find assemblage (plants from coniferous woods: n = 1,409), and they occurred in virtually
every stratigraphic unit (SE) and sample. Spruce (*Picea abies*) needles were the second most
frequent find category, ubiquitous on T3 but apparently missing on terrace T4.

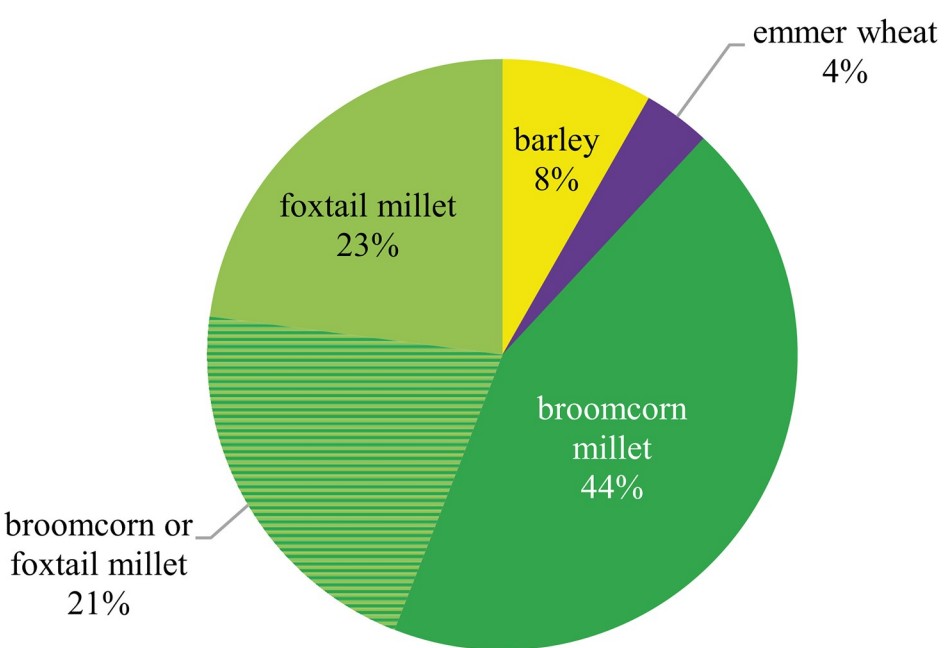

**Fig 13. Simplified spectrum of cereal grain finds at Late Bronze Age Prigglitz-Gasteil, following Stika & Heiss [177, 178] as laid out in section 2.3.4.** n = 101. Diagram: OeAW-OeAI/T. Jakobitsch.

As stated in the research goals (section 1.4), such distribution patterns as well as a general in-depth study of the wild plants at Prigglitz-Gasteil will be the focus of a follow-up publication [185] which will also incorporate palynological results as well as the detailed in-site chronology and geospatial analyses which are still under way.

## 4. Discussion

### 4.1. General observations

The large proportion (42%) of entirely unidentifiable charred remains lacking surface features is certainly the most striking characteristic of the analysed find assemblage. To some extent, this may be explained by the particular find situation of the working platforms: The samples from Prigglitz-Gasteil nearly exclusively come from cultural layers exposed to the surface and not from the protected infills of sunken features such as pits.

Furthermore, the rocky local sediment—a deposit of coarse-grained overburden and mining tailings—is much more abrasive to the fragile charred material than e. g. loess or humus-rich matrix. Consequently, prior to deposition, any "successfully" charred organic remain would be exposed to mechanical stress and multiple relocation events in a highly abrasive soil matrix, caused by trampling and surface water runoff, soil erosion, and intentional reshaping of the platforms. Exposition to erosion after deforestation could also have caused similarly high percentages of unidentified charred plant remains recovered from the Early Bronze Age contexts of Kiechlberg [17].

Compared to the other archaeobotanically investigated Alpine Bronze Age copper production sites, the observed mean find densities are among the highest (Fig 17)–except for the extraordinarily high find density reported from Mauk A which is due to local waterlogged preservation conditions. Prigglitz-Gasteil is certainly among the most intensively sampled and analysed sites.

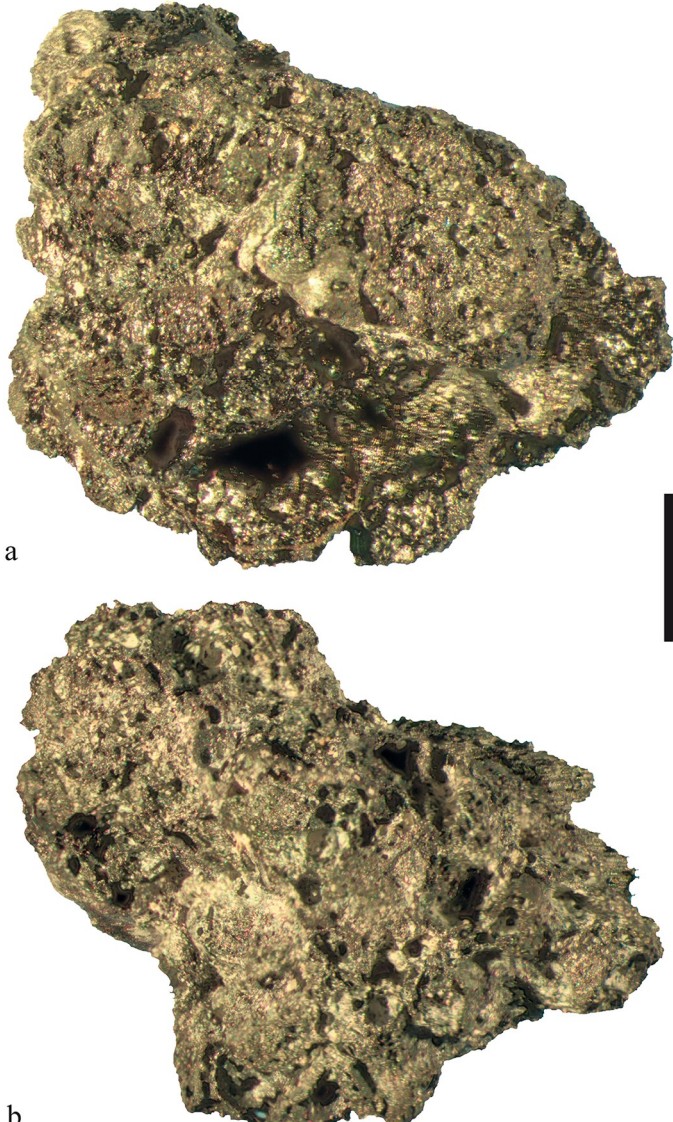

**Fig 14. Two fragments of charred cereal products from Late Bronze Age Prigglitz-Gasteil (see also Table 2).** Both contain barley (*Hordeum vulgare*). Top: find no. 0008, bottom: find no. 0042. Images: OeAW-OeAI/A. G. Heiss.

Intra-site variation of find densities is high at Prigglitz-Gasteil: While most sampled phases are poor in material—in particular concerning cultivated crops—the samples from phases T4-13E and T4-13F may even represent the charred remainders of small former stocks.

Comparison between Late Bronze Age Prigglitz-Gasteil and the other copper production sites discussed in this paper is basically problematic, as the other sites are either poor in finds or not contemporary—or both. Careful consideration of possible similarities and differences is therefore required, even for a very general look at the data (Fig 18).

Taking the two sites at Mauken for a start, the hypothesis has been brought forward that the scarcity of cultivated crops could serve as an indicator for crop production elsewhere and would therefore point towards the consumer character of a site [6, 186]. The site of Prigglitz-Gasteil resulted in amounts of cultivated and wild food plants ranging somewhere between the

**Table 2. Cereal components identified in the ACOs analysed via SEM.**

| Terrace / phase | Stratigraphical unit (SE) | Find no. | Cereal taxa / identified by... |
|---|---|---|---|
| T3-08D | 2006 | 0005 | cf. Cerealia / possible longitudinal cells |
| T3-08D | 0679 | 0945 (frag. A) | *Panicum/Setaria* / glume epidermis *Setaria italica* / glume epidermis |
| T3-08D | 0679 | 0945 (frag. B) | Cerealia, non-*Hordeum* / single-layered aleurone |
| T3-08E | 2012 | 0008 | *Hordeum vulgare* / multi-layered aleurone cf. *Hordeum vulgare* / glume fragment |
| T3-08E | 0700 | 1008 | *Hordeum vulgare* / multi-layered aleurone |
| T3-10 | 2008 | 0042 | *Hordeum vulgare* / multi-layered aleurone |
| T4-13E | 1047 | 2148 | *Hordeum vulgare* / glume epidermis |
| T4-13F | 1058 | 2150 | Cerealia, non-*Hordeum* / single-layered aleurone |
| T4-13F | 1068 | 2153 (frag. A) | *Hordeum vulgare* / double-layered transverse cells |
| T4-13F | 1068 | 2153 (frag. B) | Cerealia / bran remains including aleurone |

extremes of Klinglberg (high) and Mauken (low), which currently renders this criterion uninformative for our material.

## 4.2. Food plants

**4.2.1. Comparative crop plant spectra.** Taking the available evidence on current environmental conditions (see section 1.3.3) together with what we know about the regional climate history [187], no particular environmental restraints would affect the cultivation of any of the identified crop plants close to the metallurgical site of Prigglitz-Gasteil. The same is basically true for the sites of Kiechlberg [17, 31], Klinglberg [16, 188, 189], and Mauken [12, 27, 190].

Discussion of possible other factors influencing the identified crop spectra at Prigglitz-Gasteil and the other copper production sites requires a diachronic approach. Furthermore, as even the closest archaeobotanically analysed settlements are too far away to be directly related to Prigglitz (Fig 2), information on general tendencies in the region seemed to be more useful to refer to than individual settlements would have been. For this reason, we chose to use the semi-quantitative regional data generated from representativeness indices (RI) as brought forward by Stika and Heiss [177, 178]. Fig 19 unites these considerations as a basis for discussion, also including more recent information from Popovtschak et al. [191].

Taking the crop plant spectrum of the region "Eastern Alps and their Foreland" [177, 178] during Late Bronze Age as a reference, the most important crop plants are also present in Prigglitz-Gasteil, if at quite different proportions: barley (*Hordeum vulgare*), emmer (*Triticum dicoccum*), broomcorn millet (*Panicum miliaceum*) and foxtail millet (*Setaria italica*) as well as lentil (*Lens culinaris*).

In comparison to the other copper production sites, there is, for example, a notable difference in the importance of barley between Prigglitz and Klinglberg. However, against the background of the diachronic regional changes as reported by Stika & Heiss [177, 178], this rather seems to reflect the general trend of barley's decrease in importance between Early and Late Bronze Age than any site-specific preferences. At the same time, the large amounts of millet at Prigglitz fit together well with the late arrival of millets towards Late Bronze Age, which is by now well-documented across Europe [157, 192]. Still, it must be noted that the rather large proportion of undetermined grain fragments found at Prigglitz-Gasteil may also play a big role in distorting the results.

On the legume side, the prominence of lentil at Prigglitz-Gasteil contrasts with the inner Alpine "pea-only" sites Kiechlberg and Klinglberg. Looking at the reference data, this could just as well be related to the general increase of lentil's importance towards Late Bronze Age in the Eastern Alpine region. The differing species notwithstanding, it must be noted that the

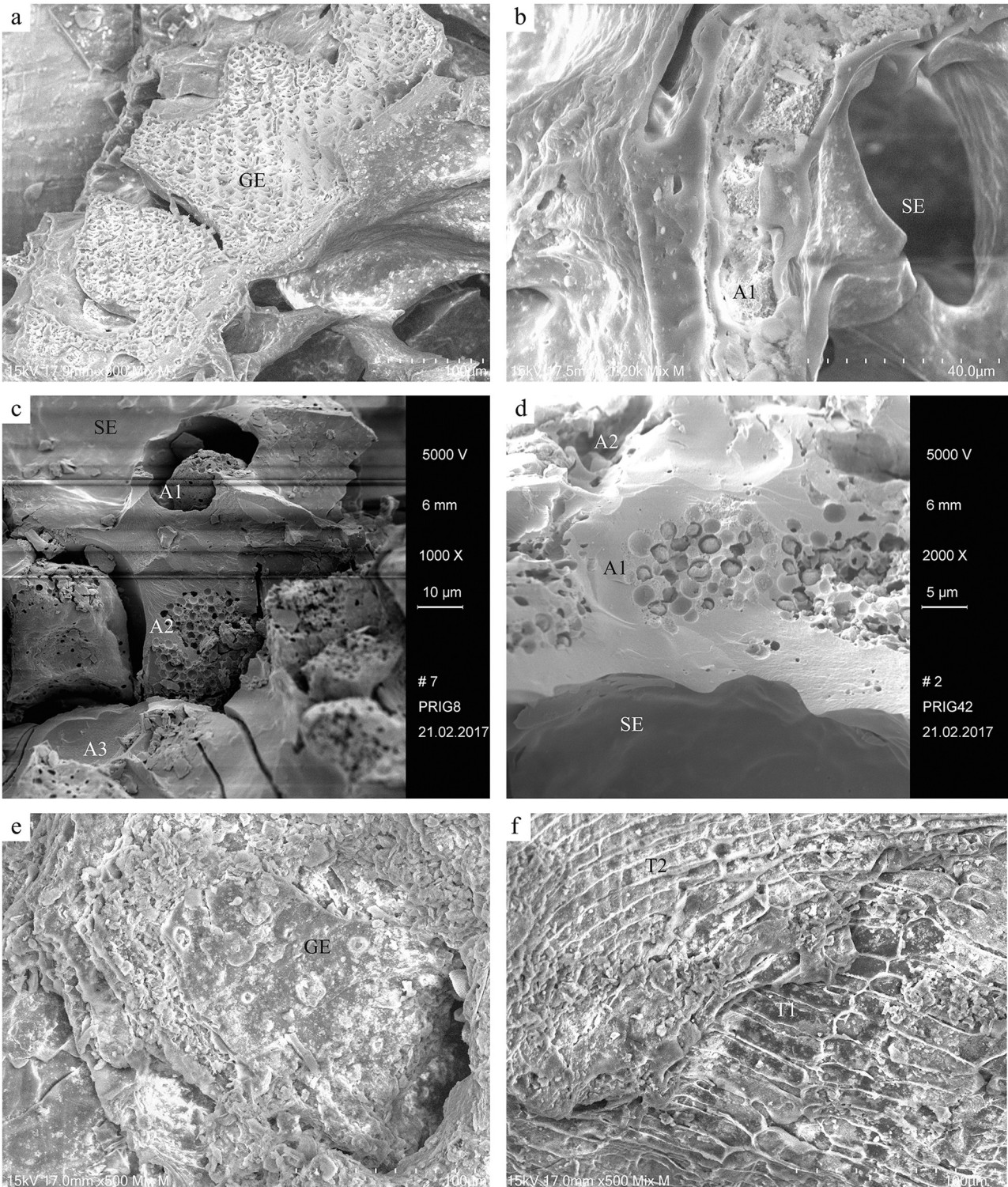

**Fig 15. Selection of SEM micrographs of the cereal products from Prigglitz-Gasteil (see also Table 2).** a) foxtail millet (*Setaria italica*) glume from find no. 0945, b) cereal (non-*Hordeum*) aleurone from find no. 0945, c) barley (*Hordeum vulgare*) aleurone from find no. 0008, d) barley (*Hordeum vulgare*) aleurone from find no. 0042, e) hulled barley (*Hordeum vulgare*) glume from find no. 2148, f) barley (*Hordeum vulgare*) transverse cells from find no. 2153. Image labels: A1, A2, A3. . . aleurone layers, GE. . . glume epidermis, SE. . . fused starchy endosperm, T1, T2. . . transverse cell layers. Images: OeAW-OeAI/A. G. Heiss.

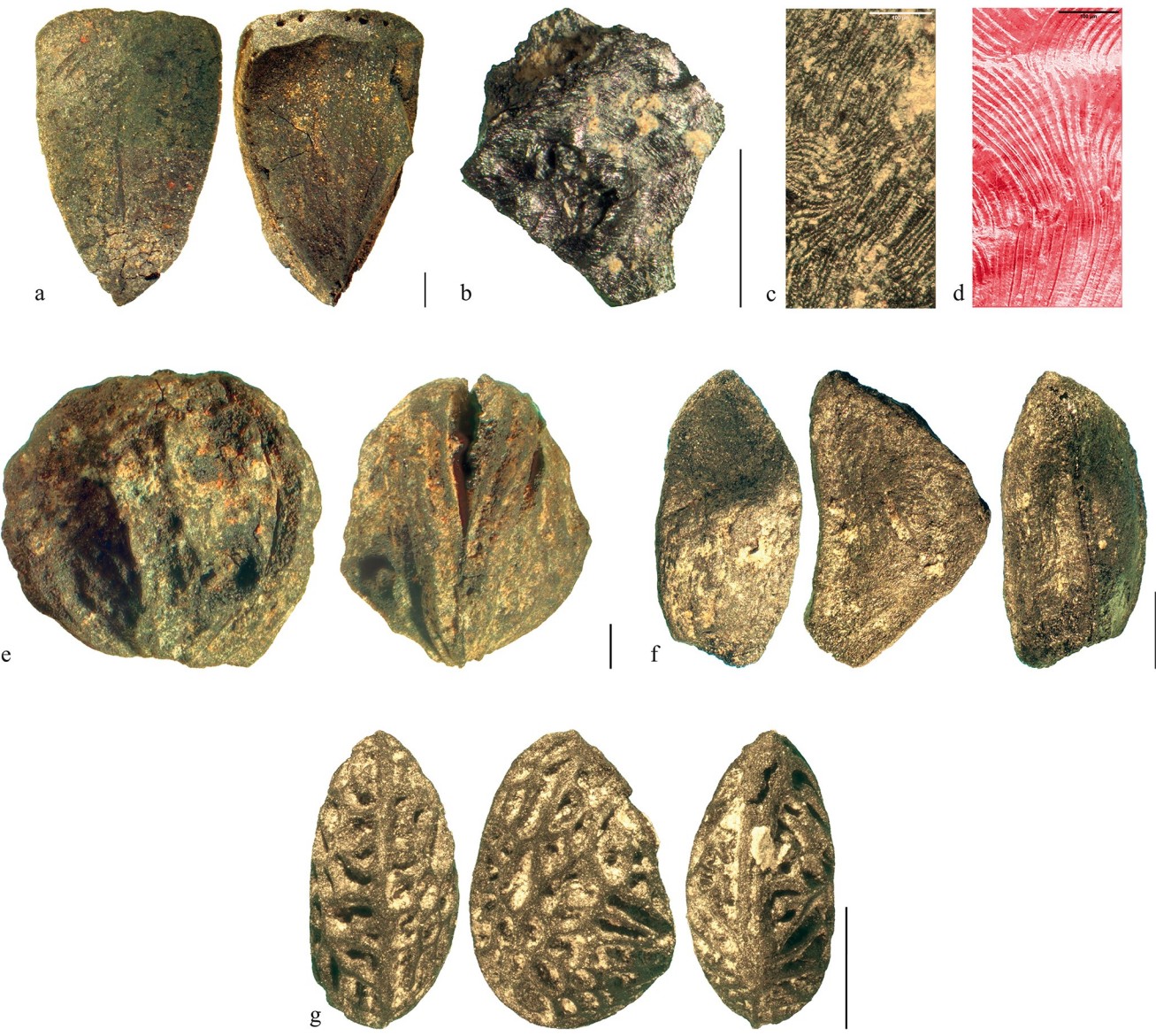

**Fig 16. Charred remains of wild edible fruit plants from Prigglitz-Gasteil.** a) hazel (*Corylus avellana*), pericarp fragment, b, c) apple/pear (*Malus*/*Pyrus* sp.) seed chamber fragment, compared to d) modern apple (*Malus domestica* 'Idared') as a reference, e) sloe (*Prunus spinosa*) stone, f) rose (*Rosa* sp.) nutlet, g) brambles (*Rubus fruticosus* agg.) stone. Scale bar lengths: 1 mm. Images: OeAW-OeAI/S. Wiesinger (a, c–g), T. Jakobitsch (b).

presence of pulses in a mining site rich in faunal evidence is an important proof that aside from the generally pork-dominated diet also plant-based protein was consumed by the local miners.

**4.2.2. Crops from field to kitchen.** As laid out in the introduction, the proportions of grain, chaff, and arable weeds in archaeobotanical find assemblages have successfully been used as indicators helping to assess the stages of cereal processing present at a site. For St. Veit-Klinglberg, S. J. Shennan and F. Green concluded from the large number of grains and the completely lacking chaff that grain processing had not been carried out on-site, and that grain supplies must have come from outside the settlement, arriving in a threshed or even—in the

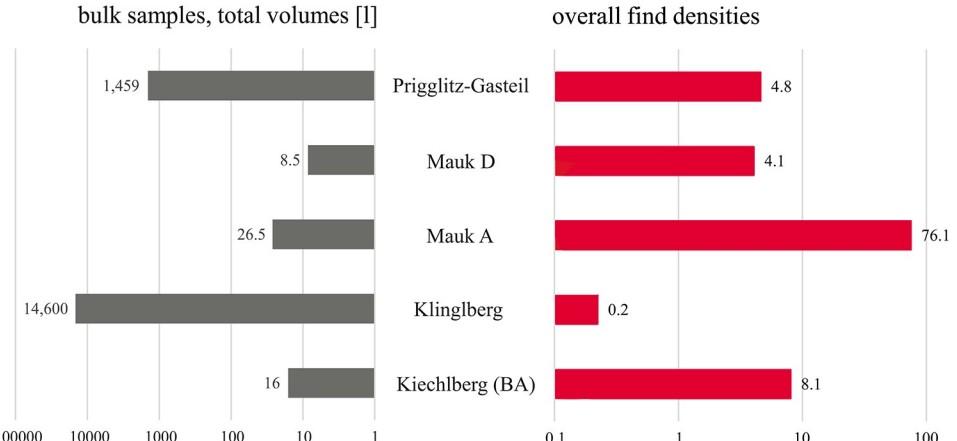

**Fig 17. Overview of the sediment samples from the sites discussed in this paper.** Left side: total sample volumes, right side: resulting find densities. Both horizontal axes are in logarithmic scale due to the differences in magnitudes. Top: younger sites, bottom: older sites. Find numbers of conifer needles from Mauken were adapted to the counting method described in section 2.3.3. For Mauk A, no figures of unidentified plant remains were available [6], thus lowering the overall find density for the site. Illustration: OeAW-OeAI/A. G. Heiss.

case of hulled wheats—dehusked state [16, 26]. The presence of a ready-made cereal product ("charred bread") was interpreted as additional strong support for this hypothesis. For the site at Kiechlberg, although mainly basing on Eneolithic finds, Schwarz & Oeggl [17] drew similar conclusions basing on the low occurrences of chaff and arable weeds.

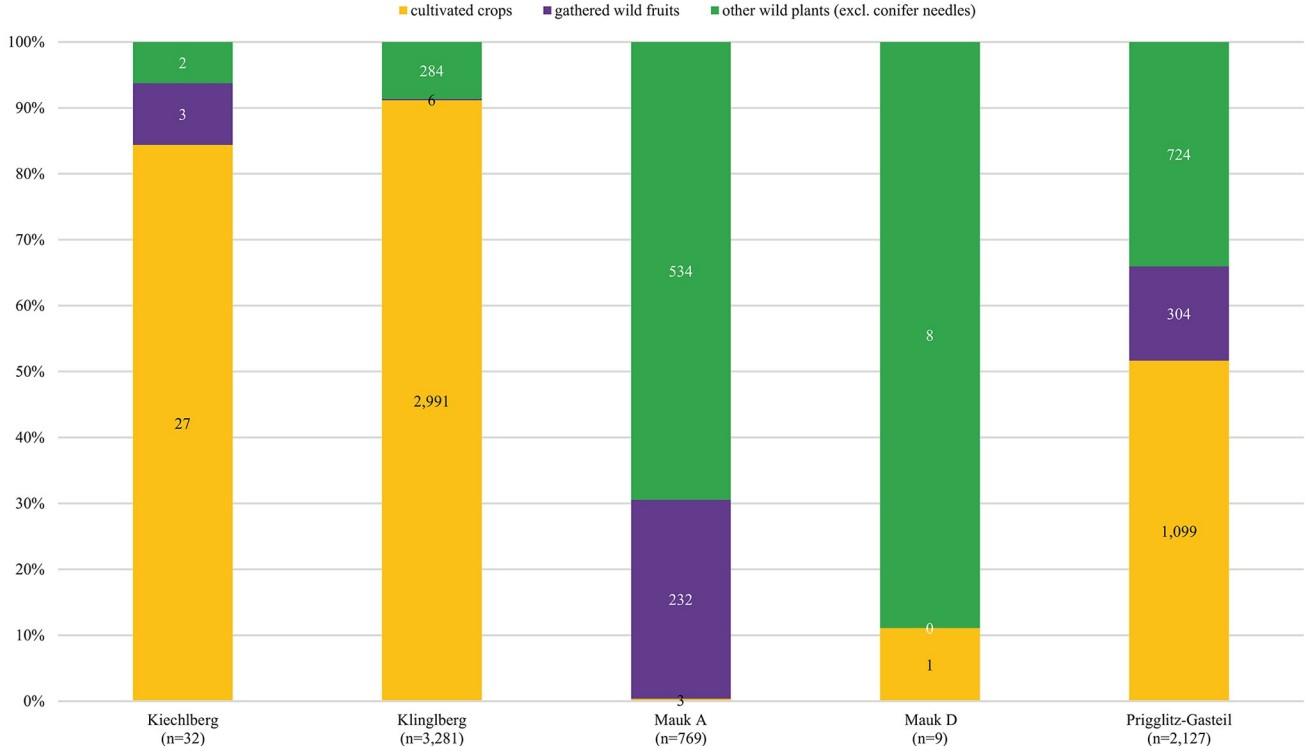

**Fig 18. Overall composition of the identifiable archaeobotanical remains (absolute counts) from the copper production sites discussed in this paper** [6, 12, 16, 17, 26, 27]. Conifer needles are excluded as "background noise". Illustration: OeAW-OeAI/A. G. Heiss.

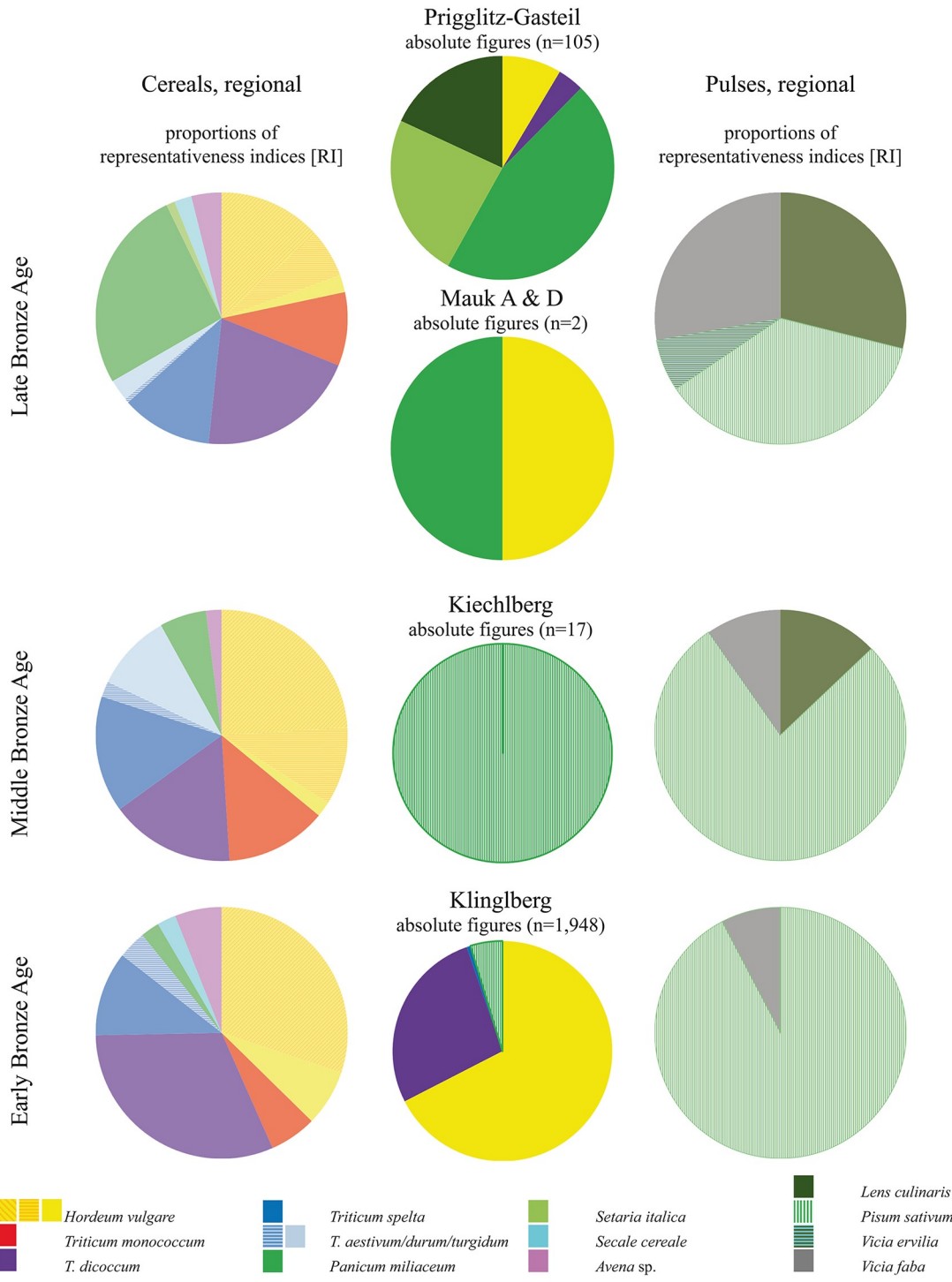

**Fig 19. Comparative crop plant spectrum from Prigglitz-Gasteil.** Cereals from Fig 3 complemented with the lentil finds from S1 Table. Central row: diachronic comparison to the results from Klinglberg [17], Kiechlberg [16, 26], and Mauken [data from 6, 12, 27]. To provide easier comparison to already published data, all count numbers were modified according to the procedures explained in section 2.3.4. Left and right rows: regional diachronic data for cereals and pulses, displayed via their respective representativeness indices (RIs) from the region "Eastern Alps and their Foreland" as published by Stika & Heiss [177, 178]. Illustration: OeAW-OeAI/A. G. Heiss.

For Prigglitz-Gasteil, we chose ternary diagrams as a familiar tool for the visualisation of these proportions. However, in contrast to the idea originally brought forward by G. E. Jones [68], we do not intend to precisely identify certain processing stages from the diagrams but merely visualize the data for the following discussion. This is the reason why in Fig 20 not only free-threshing cereals are included, but also hulled taxa.

It is important to note right away that the diagrams comparing the proportions of grain—weed—chaff (Fig 20a ans 20b, left sides) show an overall strong proportion of arable weeds in nearly all phases. However, at Prigglitz-Gasteil we do not deal with closed find contexts such as storage finds, but rather with an anthropogenically deforested, repeatedly trampled, and reshaped area (see section 4.1). Chances are therefore very high that the identified weeds are not arable weeds but that they rather reflect locally growing ruderal taxa. This gets even more likely as the metallurgical processes documented for Prigglitz-Gasteil such as smelting or alloying [40, 41] all require fire—charring will therefore only occur to a minor extent in cooking fires, while most charring events will be connected to technical fires. At the current state of research, we cannot exclude that the charred remains of weeds could be entirely unrelated to *in situ* cooking processes.

Consequently, we chose to produce an additional type of ternary plot (Fig 20a and 20b, right sides) which compares only aspects of the crop plants themselves, thereby excluding any local influences, and—more important—instead, including cereal products as the results of the "*cuisine* part" of crop processing operational sequences (see Introduction). These diagrams illustrate the nearly lacking chaff opposed to large numbers of ready-to-cook grain, and to varying amounts of processed cereal-based foodstuffs. This suggests that hypotheses on grain imported in late stages of crop cleaning (Fig 5), possibly even in a state of further processing [16, 17, 26], could very likely work just as well for Prigglitz-Gasteil. We will elaborate on this in the following section.

Due to the various conditions influencing the preservation and composition of the find assemblage, many agricultural details of the crop plants found at Prigglitz-Gasteil (see also section 4.1) are unfortunately beyond analysis, such as ecological characteristics of the crop fields [193] or the possible presence of maslins [194, 195]–in the case of Prigglitz-Gasteil, synchronous cultivation and harvest of barley and lentil in the same patches. One may be tempted to use the lack of pulses among the ACO components as an argument against such a mixed cultivation of barley and lentil. However, due to the high chance of a charred food preparation's components to "escape" analysis (see section 2.2.3), we strongly advise against any such interpretation.

**4.2.3. Comparative wild fruit spectra.**   Due to the currently uncertain origins of the weedy plants (see section 3.1.3) at Prigglitz-Gasteil, only non-weedy plants are included in the following considerations, even if his means the exclusion of potentially consumed taxa such as goosefoot [196–198] and nightshade [199].

At Prigglitz-Gasteil, gathered fruits play an important role next to (processed) cereals, their find numbers amounting to roughly a quarter of all seeds/fruits of food plants (Figs 10 and 11). Whilst this proportion is among the highest ones among the compared copper production sites (Fig 18), it is by far the most diverse one: remains of hazelnut, crab apple and/or wild pear, sloe, raspberry, dewberry, blackberry, rose, strawberry, and (black as well as red-berried) elder were found.

For the other metallurgical sites referred to in this paper, evidence of wild fruit is much more fragmentary: The Early and Middle Bronze Age layers at Kiechlberg [26], for example, delivered only three fragments of hazel (*Corylus avellana*) shells [17]. From Klinglberg, only qualitative information is published, referring to hazel shell fragments that "were recovered from a number of contexts but not in large quantities" [16], and, further on, "evidence of

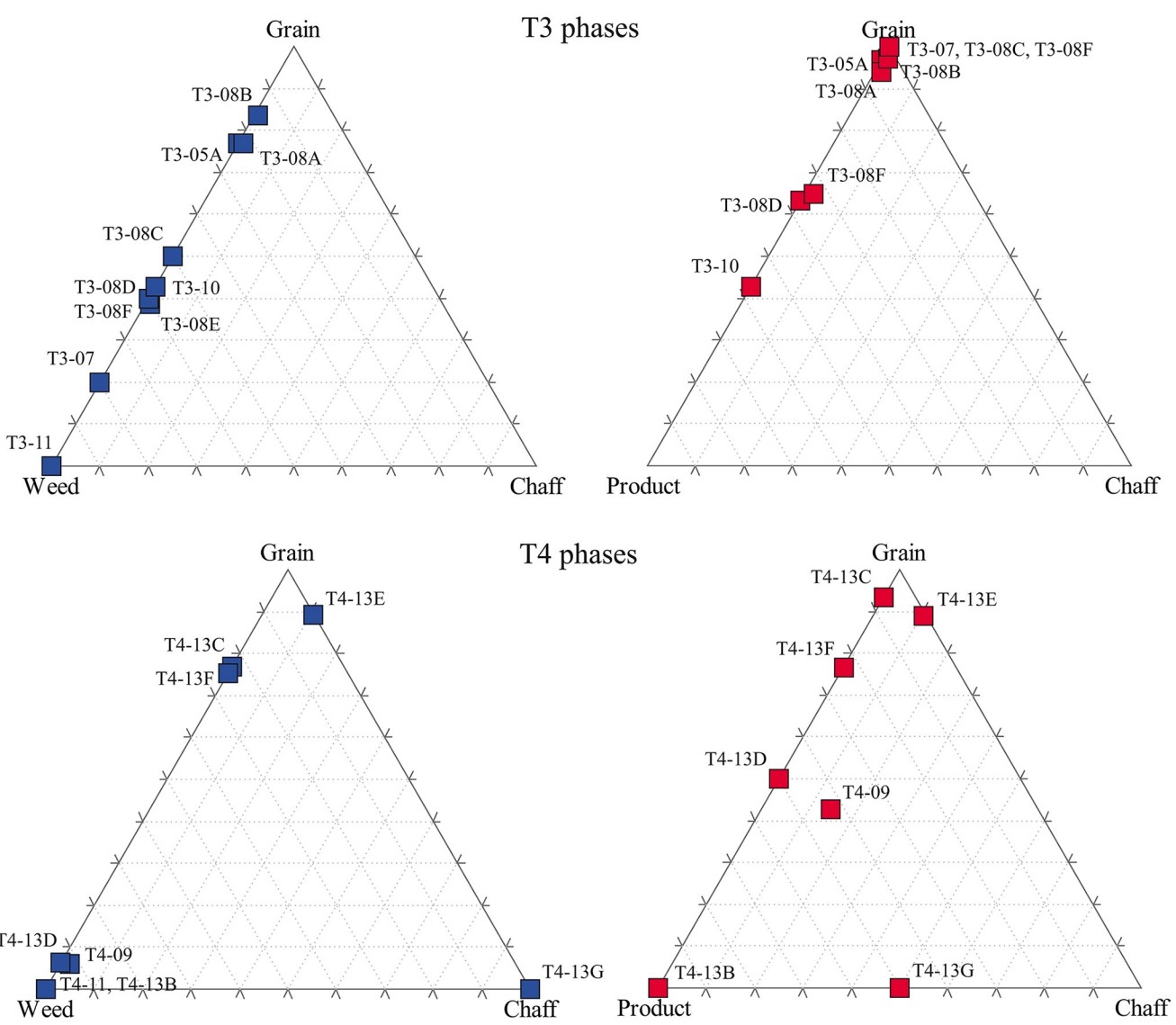

**Fig 20. Ternary plots of the find assemblages by phase, as inspired by the work of G. E. Jones [68].** a) terrace T3 (n = 696), b) terrace T4 (n = 778). Left side: proportions of grain—weed—chaff, right side: proportions of grain—product—chaff. Illustration: OeAW-OeAI/A. G. Heiss.

*Prunus* and *Rubus* species was encountered". F. Green also mentions a peculiar find of *Hippo-haë rhamnoides*, interpreted as an import [16]. At Mauk A, large numbers (> 500) of uncharred black elder (*Sambucus nigra*) fruit stones were found [6, 12], accompanied by several dozen fruit stones of raspberry (*Rubus idaeus*) and blackberry (*R. fruticosus* agg.) stones as well as a single rowan (*Sorbus aucuparia*) seed [6]–all in waterlogged preservation.

**4.2.4. Gathering the berries and nuts.** All the aforementioned taxa have in common to grow in degraded forests, on woodland margins, and on clearings [200]. When considering the available information on local vegetation composition at the respective periods and sites—Kiechlberg [17, 201], Klinglberg [202, 203], Mauken [6, 12], and Prigglitz-Gasteil (see section 1.3.3), the fruits of all these taxa were very likely easily available in the sites' immediate surroundings in the months from June until October [204]. It may seem that the specialist

communities of miners and metallurgists were not only sustained by food from agricultural production but also from foraging activities.

For Prigglitz-Gasteil, the seasonal use of such "wild" resources in the surroundings may represent a parallel to the animal find assemblage: Archaeozoological analysis suggests that either the miners themselves or other on-site craftspeople were regularly foraging the surrounding woods in spring for shed deer antlers to produce tools from them [19]. Probably they kept up the same foraging habit during summer and autumn to provide berries and nuts.

### 4.3. Plant-based food

**4.3.1. Cereal products.** Fragmented, large-seeded cereal grains do make up for the vast majority (98%) of cereal finds. Such large proportions of fragmented grains have previously been suggested as possible indicators of food processing, namely grinding or pounding, by M. van der Veen and G. Jones [56]. Exploring the potential of this large find category seemed promising, in particular because S. M. Valamoti [81] had even been able to identify pre-cooked cereal products from charred grain fragments from Greece by using the characteristic of bulging edges. She was able to document that this feature derived from an operational sequence of crushing and swelling prior to charring.

Unfortunately, neither of these considerations can be taken into account for the Prigglitz finds, as their former surfaces are usually gone due to abrasion (Fig 21). Furthermore, the few fragments with preserved fractured faces did not display any bulging edges and are therefore lacking hints on pre-charring fragmentation and possible cooking. The taphonomical considerations mentioned earlier—relocation by surface runoff, in combination with trampling—are at least as likely the reason for the high fragmentation rates of cereal grains as are actions

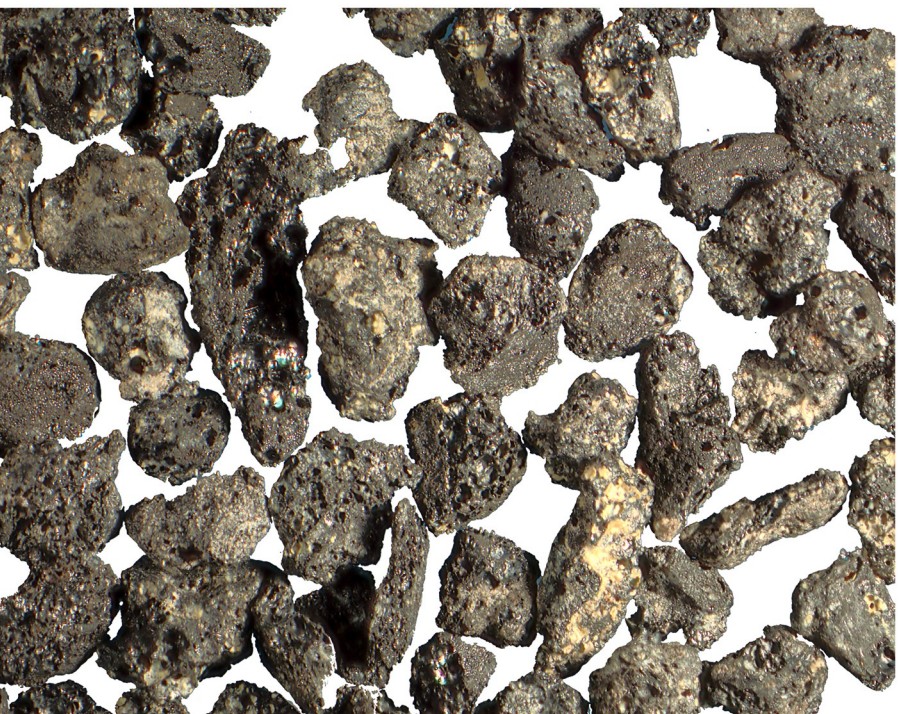

**Fig 21. Typical state of preservation of the cereal grain fragments from Prigglitz-Gasteil.** Image: OeAW-OeAI/S. Wiesinger.

related to food processing. As a temporary conclusion, we currently do not consider the degree of fragmentation of cereal grains at Prigglitz-Gasteil as informative on their artefactual character, or for the reconstruction of food-related *chaînes opératoires*.

Analysis of the cereal-based ACOs (amorphous charred objects), on the contrary, revealed a few clear hints on their production (Fig 22). Some remaining questions shall be discussed in the following (numbers corresponding to those in Fig 22).

1. **Dehusking/dehulling**: Hulled barley (*Hordeum vulgare*) was either rubbed/dehulled—thus leaving only a few accidental glume remains—or left "as is". Which of the two was the case cannot be decided due to the impossibility of quantitative statements on the ingredients of archaeological finds of processed foodstuffs. The glumes of foxtail millet (*Setaria italica*), however, had quite certainly been intended to be removed due to their limited palatability [205], but had accidentally remained [206]. No other cereals were observed, and neither were indications on other peculiarities, such as malted grains [83, 126].

2. **Separate or joint processing**: Barley and foxtail millet grains were crushed or ground for the cereal products found at Prigglitz-Gasteil. It is, however, unknown whether the two cereals were used for separate dishes or mixed at some point (Fig 22). Until now, neither were observed together in the same fragment of cereal product. If in mixture, the resulting dish could have been somehow comparable to the *Hirsotto* [193] from the contemporary settlement of Stillfried an der March (Lower Austria, see Fig 2): These charred chunks contained coarsely crushed grains of barley, broomcorn millet, and rye brome, the latter however missing from the Prigglitz material [for recipe interpretations, see 193, 207].

3. **Degree of grinding**: Too few grain fragments per cereal-based ACOs were available for measurements as to allow for any qualified statement on overall grain sizes. Consequently, no information on the degree of crushing or grinding is available for the analysed food remains.

4. **Consistency**: The preparation(s) got into contact with heat in a hydrated state as no ungelatinised starch was observed in the analysed fragments. The degree of this hydration had however not resulted in an entirely liquid mixture, as no particle size sorting was observed [83], supporting the interpretation of the remains as those of a cereal-based mush. Although water is the most likely hydrating agent, others such as milk cannot be excluded (see below).

5. **Additional ingredients**: No other additional components such as salt, condiments, fat, other cereals, etc. were observable in the material, but must be at least considered as possibilities. The chemical residue analyses on presumed cooking vessels planned for the consecutive project will hopefully shed more light on these hitherto "invisible" components.

6. **Cooking/baking**: The resulting mass was most probably not fermented due to the exclusive occurrence of micropores. Whether the final stage of preparation before charring was a cooked or raw mush is currently unknown.

With varying degrees of precision, these considerations give a general idea of **what** was produced, and **how** it was produced. The answer to the question **where** the ingredients were processed needs, however, more evidence to give clearer insights into the supply structures of Prigglitz-Gasteil and contribute to the general question of the subsistence of mining communities. Other find categories do, however, add up to the previous arguments. Firstly, finds of tools involved in food production are generally rare at the site: Of the more than 9,000 pottery fragments retrieved from the site, only around 50 show charred crusts on their surfaces,

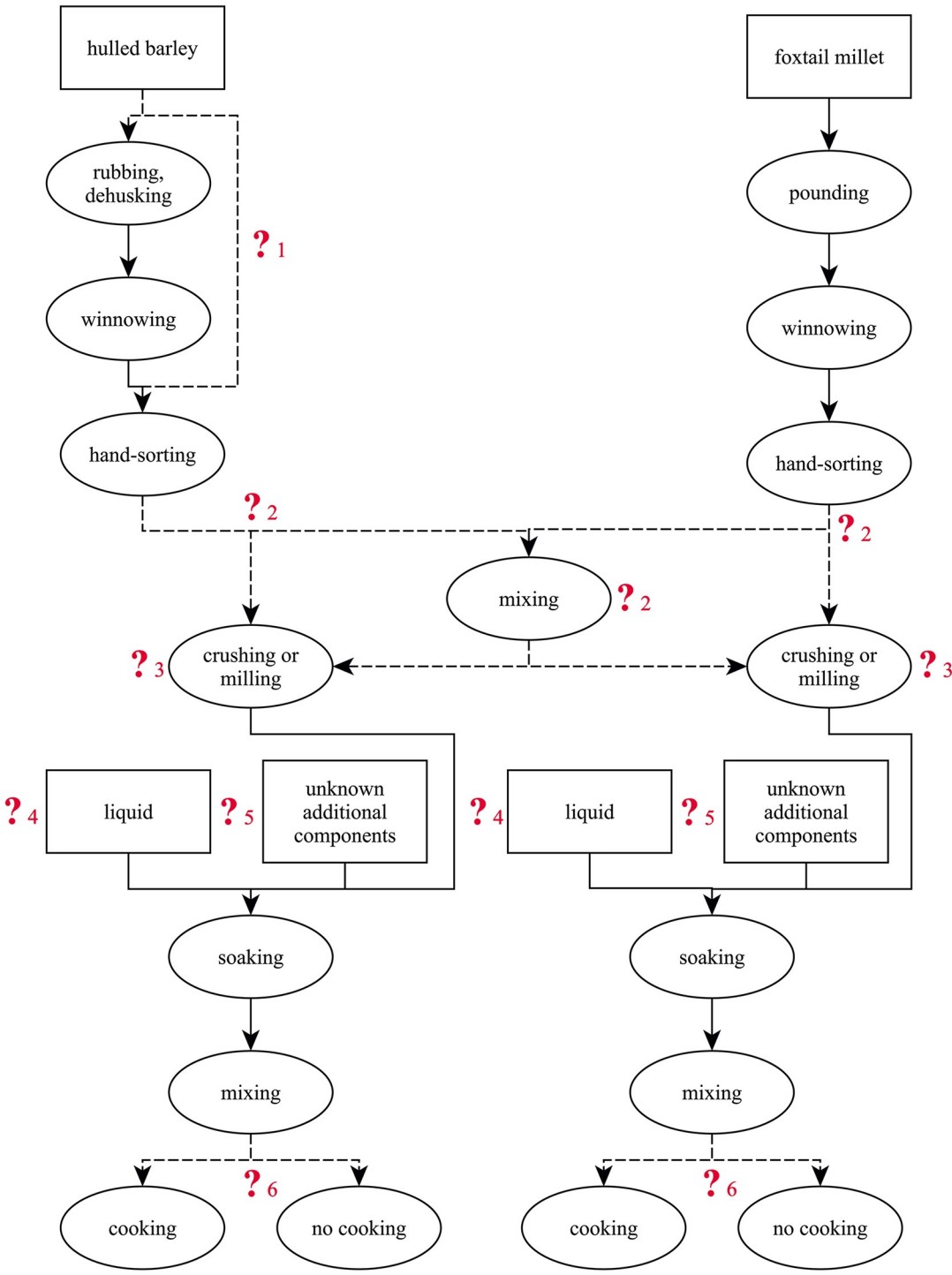

**Fig 22. Model for the *chaîne opératoire* of the charred cereal products found at Prigglitz-Gasteil.** Rectangles: components, ovals: processes. Dashed lines indicate ramifications, i. e. options/choices in processes. Numbered question marks indicate uncertainties: 1... the observed barley glumes can derive from entirely dehulled grains used "as is" or be mere remainders in intentionally dehulled barley, 2...the two cereal species were processed together or separately, 3... the grains were either finely ground, or just coarsely crushed, 4... water and/or other liquid was used for soaking; 5... other ingredients were possibly used, 6... the mushy cereal preparation was intended to be eaten either raw or cooked. Diagram: OeAW-OeAI/A. G. Heiss.

pointing towards their former use as cooking vessels [208]. Whether these were used for cooking on site or were transported to the mining site together with their pre-cooked contents, is currently uncertain. However, remains of grinding stones which represent a much more "immobile" kind of implement than pottery does, are entirely missing in the areas excavated so far [208]. It seems therefore at least highly improbable that any grinding/crushing of grains was carried out on-site.

**4.3.2. What about**. . . **processed fruits and nuts?**   We mentioned earlier that, under the premise of direct consumption in a fresh state, the archaeobotanical finds of fruits and nuts from Prigglitz-Gasteil would roughly indicate seasonal foraging/gathering activities from May until November. If we also consider food processing as a possibility, this conclusion will, however, become less straightforward.

There are indeed reasons why fruits and nuts would get processed prior to consumption instead of eating them raw. Bearing in mind that all sites except Mauk A yielded only assemblages of charred plant remains, we may focus on heat treatments here: Boiling, cooking, roasting, and drying of fruits and nuts can be highly useful for the detoxification of harmful compound as in *Sambucus* [209], or for the improvement of taste and palatability as in *Malus sylvestris* or *Prunus spinosa* [210]. Increasing shelf life by several weeks and months is another plausible reason for such heat treatments. The latter case has continuously been demonstrated by numerous finds of entire or halved fruits of charred crab apples (*Malus sylvestris*) from Neolithic [211–213] and Bronze Age [214–216] lakeshore settlements—and fewer finds from dryland sites [e. g. 217]–suggesting a common habit of drying them for preservation. Roasting of hazelnuts has likewise been postulated, albeit mostly for Mesolithic populations [218].

Such habits of roasting or drying fruits and nuts may indeed have led to the preservation of the numerous charred fruit fragments of crab apples and rosehips found at Prigglitz-Gasteil, and to the finds of hazelnut shell fragments from the other sites. However, interpretation of the finds of wild taxa suffers from the same possible biases as the weedy plant assemblages (see sections 3.1.3 and 4.2.3): They could simply have grown at or around the sites, and they could have been charred by sheer accident in the technical fires, or intentionally burned as waste (see also Green's considerations [16]).

As a conclusion for this section, while pointing out the possibility of processed fruits/nut, it must be clearly stated that none of the charred wild fruit finds from either Prigglitz-Gasteil or the other sites allow unequivocal conclusions on cooking processes, neither do they allow clear implications on the seasonality of the sites concerned.

## 5. Conclusions and outlook

Prigglitz-Gasteil is among the most intensively sampled and most species-rich Bronze Age mining sites in the Eastern Alps. The utilised crop plant spectrum lies within the range expected for the period and region. From the cultivated taxa, only barley and foxtail millet are documented as components of the analysed cereal preparations—either processed in mixture or separately, these two could have made up the major components of a miners' dish, a simple unfermented cereal mush. The role of pulses and wild fruits/nuts in Bronze Age "mining *cuisine*", however, still has to remain vague.

Analysis of more cereal product fragments from Prigglitz-Gasteil, together with a re-evaluation of the "charred bread" from Klinglberg via SEM, will help clarify the currently uncertain aspects of their components and production, accompanied by chemical residue analysis of the supposed cooking vessels from Prigglitz-Gasteil. The very encouraging results from Stillfried/March [120, 193], a settlement in Lower Austria contemporary to Prigglitz-Gasteil, even make

it conceivable for Prigglitz-Gasteil to address potentially existing culinary variability within the same site.

At Prigglitz-Gasteil, cereals were likely brought from outside in the form of ready-to-cook grains and ground flour/meal to sustain the workers with food ingredients in order to cook them on site. Some food may even have been delivered in pre-cooked state. This general impression confirms previous interpretations of archaeobotanically investigated Bronze Age metallurgical sites.

Where exactly the agricultural production took place is a question that will need clarification in the future. Adjacent farmsteads could have provided the mining site with food resources, but also more remote settlements could have contributed. At the same time, foraging for natural resources probably played a significant role in Prigglitz-Gasteil, partially fulfilling the need for raw materials (shed antlers) and for nutritional supplements (gathered wild fruit) taken from the natural surroundings.

The distribution patterns of plant finds indicate functional intra-site differentiation (e. g. a possible small cereal stock in SE 1047; plant spectra differing between T3 and T4; weedy plants distribution as contrasted to cereals distribution) and might even derive from entirely different processes happening on the two working terraces. Analysis of the remaining archaeobotanical samples from the late Bronze Age layers, together with high-resolution spatial and diachronic evaluation of other small finds (charcoal, casting droplets, and antler and bone fragments) will hopefully help render the picture of metallurgy and other craftsmanship at Prigglitz-Gasteil more complete, and will result in better insights into the use of space at the site. Supplies of the mining and metallurgical operations with construction timber and firewood are currently under investigation. The results will be presented in a follow-up publication [185], and will be accompanied by palynological data on local vegetation history.

## Supporting information

**S1 Table. The charred plant remains from the Late Bronze Age layers of Prigglitz-Gasteil.** Sheet 1: counts by individual samples, sheet 2: counts summed up to phases. Red squares: Cereal-based ACOs analysed via SEM. Data: OeAW-OeAI/T. Jakobitsch, S. Wiesinger, A. G. Heiss.
(XLSX)

## Acknowledgments

We thank Ilona Szunyogh (University of Natural Resources and Life Sciences, Vienna— BOKU), Michael Konrad and Julia Längauer (both Danube University Krems—DUK) for the meticulous flotation of the samples (I. Szunyogh, M. Konrad), and for scanning through the heavy fractions (I. Szunyogh, J. Längauer). We are grateful to Marianne Kohler-Schneider (BOKU) who kindly supported the first author in scientific and organisational issues during the pilot project. The authors warmly thank Klaus Oeggl (University of Innsbruck) for supporting us with background information on the Kiechlberg material, and for his cooperation to conduct palynological investigations in the area. Our thanks also go to Erika Rücker and Anne Heller (both University of Hohenheim) for their help in producing the PlantCult-funded SEM images. We thank Soultana Maria Valamoti (Aristotle University of Thessaloniki), PI of ERC Project PlantCult, for all the constructive exchange and for the great time we had in the project. For their helpful suggestions during review, we thank Liliana Janik (University of Cambridge) and Kerstin Kowarik (Natural History Museum Vienna).

## Author Contributions

**Conceptualization:** Andreas G. Heiss, Thorsten Jakobitsch, Silvia Wiesinger, Peter Trebsche.

**Data curation:** Andreas G. Heiss, Thorsten Jakobitsch, Silvia Wiesinger.

**Formal analysis:** Andreas G. Heiss, Thorsten Jakobitsch, Silvia Wiesinger.

**Funding acquisition:** Peter Trebsche.

**Investigation:** Andreas G. Heiss, Thorsten Jakobitsch, Silvia Wiesinger.

**Methodology:** Andreas G. Heiss, Thorsten Jakobitsch, Peter Trebsche.

**Project administration:** Andreas G. Heiss, Peter Trebsche.

**Resources:** Andreas G. Heiss.

**Supervision:** Andreas G. Heiss, Peter Trebsche.

**Validation:** Andreas G. Heiss, Thorsten Jakobitsch.

**Visualization:** Andreas G. Heiss, Thorsten Jakobitsch, Silvia Wiesinger.

**Writing – original draft:** Andreas G. Heiss, Thorsten Jakobitsch.

**Writing – review & editing:** Andreas G. Heiss, Thorsten Jakobitsch, Silvia Wiesinger, Peter Trebsche.

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
