## [Decision Letter · Decision Letter 0]

5 Nov 2020

PONE-D-20-28652

Dig Out, Dig In! The plant-based diet of Late Bronze Age miners at the copper production site of Prigglitz-Gasteil (Lower Austria), and a few general thoughts on archaeological remains of processed food

PLOS ONE

Dear Dr. Heiss,

Thank you for submitting your manuscript to PLOS ONE. After careful consideration, we feel that it has merit but does not fully meet PLOS ONE’s publication criteria as it currently stands. Therefore, we invite you to submit a revised version of the manuscript that addresses the points raised during the review process.

All comments must be fully addressed before re-submission.

We look forward to receiving your revised manuscript.

Kind regards,

Peter F. Biehl, PhD

Academic Editor

PLOS ONE

Additional Editor Comments:

Your manuscript has now been seen by a referee, whose comments are appended below. You will see from these comments that while the referees find your work of potential interest, it has raised substantial concerns that must be addressed. In light of these comments, we cannot accept the manuscript for publication, but would be interested in considering a revised version that addresses these serious concerns.

Journal Requirements:

2. We note that Figures 2 and 3 in your submission contain map/satellite images which may be copyrighted. All PLOS content is published under the Creative Commons Attribution License (CC BY 4.0), which means that the manuscript, images, and Supporting Information files will be freely available online, and any third party is permitted to access, download, copy, distribute, and use these materials in any way, even commercially, with proper attribution. For these reasons, we cannot publish previously copyrighted maps or satellite images created using proprietary data, such as Google software (Google Maps, Street View, and Earth). For more information, see our copyright guidelines: http://journals.plos.org/plosone/s/licenses-and-copyright.

2.1.    You may seek permission from the original copyright holder of Figures 2 and 3 to publish the content specifically under the CC BY 4.0 license. 

2.2.    If you are unable to obtain permission from the original copyright holder to publish these figures under the CC BY 4.0 license or if the copyright holder’s requirements are incompatible with the CC BY 4.0 license, please either i) remove the figure or ii) supply a replacement figure that complies with the CC BY 4.0 license. Please check copyright information on all replacement figures and update the figure caption with source information. If applicable, please specify in the figure caption text when a figure is similar but not identical to the original image and is therefore for illustrative purposes only.

3. We note that Figure 7 includes an image of a  participant in the study. 

Reviewers' comments:

Reviewer's Responses to Questions

**Comments to the Author**

1. Is the manuscript technically sound, and do the data support the conclusions?

Reviewer #1: Partly

2. Has the statistical analysis been performed appropriately and rigorously? 

Reviewer #1: Yes

3. Have the authors made all data underlying the findings in their manuscript fully available?

Reviewer #1: Yes

4. Is the manuscript presented in an intelligible fashion and written in standard English?

Reviewer #1: Yes

5. Review Comments to the Author

Reviewer #1: It is an interesting article and worth publishing, with valuable new data that can enhance our knowledge and interpretation of the past

Before publishing, I would recommend restructuring the article’s focus from just a report on the findings of a research project, to how the research project findings contribute further to our understanding of Late Bronze Age economy, societies etc. Hence, I suggest the paper is not just about the Prigglitz-Gasteil site, but rather a ‘window’ into the economy and food supply to work forces linked with mining in Late Bronze Age societies of Eastern Alps, Lower Austria, Styria (where it is?- it is a regional name the majority of readers might not be familiar with) and Western Hungary – page 4

That would alter the text e.g. by deleting or modifying the text on vegetation history of the site or proximity of other Late Bronze Age sites: if such information is not used in the interpretation of the findings, maybe it is redundant? - page 8

The Abstract and Introduction do not reflect the ideas presented in the article, including cuisine, which goes beyond the idea of consumption. Further, the text cannot only focus on the description of findings without linking it into the concepts of cuisine as proposed in the first part of the article.

I am not sure how the concept of cuisine is relevant to question posed in this paper, since the question are related to the food provision and consumption.

1) The author needs to make clear that he looks only at the charred material and a small amount of parenchyma. Such remains are very valuable when talking about cuisine if the data is related to other types of plant remains as phytoliths (Saul H, Madella M, Fischer A, Glykou A, Hartz S, et al. (2013) Phytoliths in Pottery Reveal the Use of Spice in European Prehistoric Cuisine. PLoS ONE 8(8): e70583.doi:10.1371journal.pone.0070583) or protein (Hendy, J., Colonese, A.C., Franz, I. et al. Ancient proteins from ceramic vessels at Çatalhöyük West reveal the hidden cuisine of early farmers. Nat Commun 9, 4064 (2018). https://doi.org/10.1038/s41467-018-06335-6).

2) It would be useful if the author could acknowledge that he is presenting only part of the picture of what is considered the chain operatoire of a cuisine, since he does not talk about ways of food preparation like baking, cooking, eating raw, or ways of consumption. Fig 22 is not sufficient to answer those questions.

3) There is no discussion on the role of wild plants, e.g. nuts or fruits, in later Bronze Age cuisine.

4) There is no discussion about the legumes as part of obtaining, preparing and consumption of food stuffs.

5) Since the focus of the article is related mainly to the charred materia,l I suggest looking at the food as material, as in plant food consumption e.g. Van der Venn, M. 2008. Food as embodied material culture: diversity and change in plant food consumption. Journal of Roman Archaeology 21: 83-109. This would ground the author in the issue of plants as material culture.

The part where the data is presented is well structured and written with very good figures illustrating the points made.

In the discussion, I would suggest returning to the idea of cuisine (if it is a concept the author decides to keep), and answer the questions based on the charred and parenchyma data from the Prigglitz-Gasteil site. For example, if and how if so, the diet of the miners was restricted due to the food provisions coming from outside the site? What does the presence of legumes tell us about food production? Were they cultivated in similar a way to cereals or were they grown among cereal crops, and hence appeared in assemblages?

The wild plant remains are treated in the text as auxiliary data. Their availability during particular seasons, like elder or strawberry, indicate the part of the year work in the mines took place. Moreover, other plants like e.g. sloe, hazel, or wild fruits, indicate other seasons or storage practices as part of the food supply strategy. It is assumed in the text Last but not least: fruits and nuts that wild fruits and nuts were foraged, but by whom, the miners or food suppliers?

Were ‘The most frequent weed taxa belong to members of the goosefoot family (Chenopodiaceae=Amaranthaceae p. p.) and to wild millets (Poaceae-Panicoideae, including e. g. Echinochloa and wild Setaria taxa). These wild millets showed an overall distribution pattern which seemed similar to the one of cultivated millets’ (p22) intentionally gathered? If so what this tell us about food preferences, food supply etc? Further, what does the consumption of cultivated and wild food tells us about Late Bronze Age economy in terms of food production and labour specialisation.

6. PLOS authors have the option to publish the peer review history of their article (what does this mean?). If published, this will include your full peer review and any attached files.

Reviewer #1: **Yes: **Liliana Janik

---

## [Author Response · Author response to Decision Letter 0]

30 Jan 2021

PONE-D-20-28652

Dig Out, Dig In! The plant-based diet of Late Bronze Age miners at the copper production site of Prigglitz-Gasteil (Lower Austria), and a few general thoughts on archaeological remains of processed food

PLOS ONE

Dear Dr. Heiss,

Thank you for submitting your manuscript to PLOS ONE. After careful consideration, we feel that it has merit but does not fully meet PLOS ONE’s publication criteria as it currently stands. Therefore, we invite you to submit a revised version of the manuscript that addresses the points raised during the review process.

All comments must be fully addressed before re-submission.

We look forward to receiving your revised manuscript.

Kind regards,

Peter F. Biehl, PhD

Academic Editor

PLOS ONE

Additional Editor Comments:

Your manuscript has now been seen by a referee, whose comments are appended below. You will see from these comments that while the referees find your work of potential interest, it has raised substantial concerns that must be addressed. In light of these comments, we cannot accept the manuscript for publication, but would be interested in considering a revised version that addresses these serious concerns.

Journal Requirements:

-> We have carefully re-checked the requirements and hope that our manuscript meets PLOS One’s style requirements.

2. We note that Figures 2 and 3 in your submission contain map/satellite images which may be copyrighted. All PLOS content is published under the Creative Commons Attribution License (CC BY 4.0), which means that the manuscript, images, and Supporting Information files will be freely available online, and any third party is permitted to access, download, copy, distribute, and use these materials in any way, even commercially, with proper attribution. For these reasons, we cannot publish previously copyrighted maps or satellite images created using proprietary data, such as Google software (Google Maps, Street View, and Earth). For more information, see our copyright guidelines: http://journals.plos.org/plosone/s/licenses-and-copyright.

2.1. You may seek permission from the original copyright holder of Figures 2 and 3 to publish the content specifically under the CC BY 4.0 license. 

2.2. If you are unable to obtain permission from the original copyright holder to publish these figures under the CC BY 4.0 license or if the copyright holder’s requirements are incompatible with the CC BY 4.0 license, please either i) remove the figure or ii) supply a replacement figure that complies with the CC BY 4.0 license. Please check copyright information on all replacement figures and update the figure caption with source information. If applicable, please specify in the figure caption text when a figure is similar but not identical to the original image and is therefore for illustrative purposes only.

-> The map in Figure 2 was made by the first author using ArcGIS Pro. This fact, alongside with the underlying data, is now explicitly stated in the Methods section.

-> The aerial photograph in Figure 3 is freely publishable under a CC-BY license. The content permission is now added to the submission.

3. We note that Figure 7 includes an image of a participant in the study. 

-> We had already obtained informed consent from the person depicted in Figure 7, and the consent form had been properly signed and submitted with the manuscript. As the individual is not a patient but a colleague, there is no such thing as case notes. We are therefore unsure what else to do with the consent form but resubmit it.

-> We have added the suggested statement to the Methods section in an adapted form.

Reviewers' comments:

Reviewer's Responses to Questions

Comments to the Author

1. Is the manuscript technically sound, and do the data support the conclusions?

Reviewer #1: Partly

2. Has the statistical analysis been performed appropriately and rigorously? 

Reviewer #1: Yes

3. Have the authors made all data underlying the findings in their manuscript fully available?

Reviewer #1: Yes

4. Is the manuscript presented in an intelligible fashion and written in standard English?

Reviewer #1: Yes

5. Review Comments to the Author

Reviewer #1: It is an interesting article and worth publishing, with valuable new data that can enhance our knowledge and interpretation of the past

Before publishing, I would recommend restructuring the article’s focus from just a report on the findings of a research project, to how the research project findings contribute further to our understanding of Late Bronze Age economy, societies etc. Hence, I suggest the paper is not just about the Prigglitz-Gasteil site, but rather a ‘window’ into the economy and food supply to work forces linked with mining in Late Bronze Age societies of Eastern Alps, Lower Austria, Styria (where it is?- it is a regional name the majority of readers might not be familiar with) and Western Hungary – page 4

-> Taking these considerations into account, together with comment #5, we have now improved the section “Research goals” to make our intentions more visible, and to present the scope and limitations of the current study. Moreover, the entire manuscript structure has been reorganised. We hope that insufficiently explained thoughts are now more clearly laid out to the reader, and that the overarching questions are now addressed in a better way.

-> Toponyms have been amended accordingly. 

That would alter the text e.g. by deleting or modifying the text on vegetation history of the site or proximity of other Late Bronze Age sites: if such information is not used in the interpretation of the findings, maybe it is redundant? - page 8

-> We are now referring to local vegetation in the Discussion section, and are generally elaborating on gathered fruit in much greater detail.

The Abstract and Introduction do not reflect the ideas presented in the article, including cuisine, which goes beyond the idea of consumption. Further, the text cannot only focus on the description of findings without linking it into the concepts of cuisine as proposed in the first part of the article.

-> We have adjusted the manuscript’s title, abstract, and introduction accordingly.

-> The discussion part has been amended accordingly.

I am not sure how the concept of cuisine is relevant to question posed in this paper, since the question are related to the food provision and consumption.

-> We think that it makes a difference to include culinary aspects into the chaîne opératoire of cereal processing: The information which elements of the various processing stages from harvested sheaves/grain to ready-to-eat dishes allows to gain information on supply chains as it has not been accessible before.

-> We have now tried to explain this in a better way, and to better highlight cuisine as the “missing link” between crop and consumption.

1) The author needs to make clear that he looks only at the charred material and a small amount of parenchyma. Such remains are very valuable when talking about cuisine if the data is related to other types of plant remains as phytoliths (Saul H, Madella M, Fischer A, Glykou A, Hartz S, et al. (2013) Phytoliths in Pottery Reveal the Use of Spice in European Prehistoric Cuisine. PLoS ONE 8(8): e70583.doi:10.1371journal.pone.0070583) or protein (Hendy, J., Colonese, A.C., Franz, I. et al. Ancient proteins from ceramic vessels at Çatalhöyük West reveal the hidden cuisine of early farmers. Nat Commun 9, 4064 (2018). https://doi.org/10.1038/s41467-018-06335-6).

-> We have looked at the total assemblage of charred plant macroremains and amorphous charred objects (ACO); all ACOs larger than 2 mm were evaluated for any preserved plant tissues, resulting in their classification as cereal products and fruit parenchyma. We hope that now we have been able to explain these issues in a better way.

-> We are aware of the fact that additional remains will bring additional information, but see that this has not been explained very well. We are aware of the possibilities in proteomics, aDNA, and lipid residual analyses, however, these analyses are within the scope of the follow-up projects. We have now tried to better explain this.

-> The identified charred silica skeletons of e. g. the millet glumes are indeed phytoliths, although we did not explicitly use this term.

2) It would be useful if the author could acknowledge that he is presenting only part of the picture of what is considered the chain operatoire of a cuisine, since he does not talk about ways of food preparation like baking, cooking, eating raw, or ways of consumption. Fig 22 is not sufficient to answer those questions.

-> We have now elaborated on the issues the reviewer has raised, and hope that now we explain them sufficiently.

3) There is no discussion on the role of wild plants, e.g. nuts or fruits, in later Bronze Age cuisine.

-> We have added discussion on these aspects.

4) There is no discussion about the legumes as part of obtaining, preparing and consumption of food stuffs.

-> We have added discussion on these aspects.

5) Since the focus of the article is related mainly to the charred materia,l I suggest looking at the food as material, as in plant food consumption e.g. Van der Venn, M. 2008. Food as embodied material culture: diversity and change in plant food consumption. Journal of Roman Archaeology 21: 83-109. This would ground the author in the issue of plants as material culture.

-> We realise that we had taken for granted that plants from archaeological contexts would be considered a part of material culture, and instead immediately focused on less clear issues such as the differentiation between their “ecofacts state” (grains/seeds) and their “artefacts state” (processed foodstuffs, dishes). We have now modified the manuscript accordingly, in order to better explain our intentions.

-> The reference to Marijke van der Veen’s paper has been added.

The part where the data is presented is well structured and written with very good figures illustrating the points made.

In the discussion, I would suggest returning to the idea of cuisine (if it is a concept the author decides to keep), and answer the questions based on the charred and parenchyma data from the Prigglitz-Gasteil site. For example, if and how if so, the diet of the miners was restricted due to the food provisions coming from outside the site? What does the presence of legumes tell us about food production? Were they cultivated in similar a way to cereals or were they grown among cereal crops, and hence appeared in assemblages?

-> We have now elaborated on all issues raised in the discussion section within the course of the general restructuring of the manuscript.

The wild plant remains are treated in the text as auxiliary data. Their availability during particular seasons, like elder or strawberry, indicate the part of the year work in the mines took place. Moreover, other plants like e.g. sloe, hazel, or wild fruits, indicate other seasons or storage practices as part of the food supply strategy. It is assumed in the text Last but not least: fruits and nuts that wild fruits and nuts were foraged, but by whom, the miners or food suppliers?

Were ‘The most frequent weed taxa belong to members of the goosefoot family (Chenopodiaceae=Amaranthaceae p. p.) and to wild millets (Poaceae-Panicoideae, including e. g. Echinochloa and wild Setaria taxa). These wild millets showed an overall distribution pattern which seemed similar to the one of cultivated millets’ (p22) intentionally gathered? If so what this tell us about food preferences, food supply etc? Further, what does the consumption of cultivated and wild food tells us about Late Bronze Age economy in terms of food production and labour specialisation.

-> No evidence currently speaks against the miners/metallurgists or other “sur place” craftspeople gathering the wild fruits as well as the antlers in the surrounding woods themselves. We have now added discussion of these considerations to the manuscript.

-> The overlapping distribution patterns of cultivated and wild millets could possibly indicate their origin from the same fields, and thus a weedy character of the wild millets. The Amaranthaceae p. p. show an inverse pattern (at least when comparing T3 vs. T4). At the current state of analysis, we prefer not to interpret these weedy taxa, as the fine chronology is still missing: Although the overall temporal range of LBA activities is narrow in Prigglitz, the distribution patterns also contain depth in time. A forthcoming publication will deal with the distribution patterns of wild and cultivated species and help clear up these questions.

-> We have now pointed out these issues and considerations in the manuscript.

6. PLOS authors have the option to publish the peer review history of their article (what does this mean?). If published, this will include your full peer review and any attached files.

Do you want your identity to be public for this peer review? For information about this choice, including consent withdrawal, please see our Privacy Policy.

Reviewer #1: Yes: Liliana Janik

---

## [Decision Letter · Decision Letter 1]

24 Feb 2021

Dig Out, Dig In! Plant-based diet at the Late Bronze Age copper production site of Prigglitz-Gasteil (Lower Austria) and the relevance of processed foodstuffs for the supply of Alpine Bronze Age miners

PONE-D-20-28652R1

Dear Dr. Heiss,

We’re pleased to inform you that your manuscript has been judged scientifically suitable for publication and will be formally accepted for publication once it meets all outstanding technical requirements.

Kind regards,

Peter F. Biehl, PhD

Academic Editor

PLOS ONE

Additional Editor Comments (optional):

Reviewers' comments:

Reviewer's Responses to Questions

**Comments to the Author**

1. If the authors have adequately addressed your comments raised in a previous round of review and you feel that this manuscript is now acceptable for publication, you may indicate that here to bypass the “Comments to the Author” section, enter your conflict of interest statement in the “Confidential to Editor” section, and submit your "Accept" recommendation.

Reviewer #1: All comments have been addressed

2. Is the manuscript technically sound, and do the data support the conclusions?

Reviewer #1: Yes

3. Has the statistical analysis been performed appropriately and rigorously? 

Reviewer #1: N/A

4. Have the authors made all data underlying the findings in their manuscript fully available?

Reviewer #1: Yes

5. Is the manuscript presented in an intelligible fashion and written in standard English?

Reviewer #1: Yes

6. Review Comments to the Author

Reviewer #1: All questions have been fully answered and suggestions considered. I am satisfied withe the quality of the text and illustrations

7. PLOS authors have the option to publish the peer review history of their article (what does this mean?). If published, this will include your full peer review and any attached files.

Reviewer #1: **Yes: **Dr Liliana Janik

---

## [Editor Report · Acceptance letter]

1 Mar 2021

PONE-D-20-28652R1 

Dig Out, Dig In! Plant-based diet at the Late Bronze Age copper production site of Prigglitz-Gasteil (Lower Austria) and the relevance of processed foodstuffs for the supply of Alpine Bronze Age miners 

Dear Dr. Heiss:

I'm pleased to inform you that your manuscript has been deemed suitable for publication in PLOS ONE. Congratulations! Your manuscript is now with our production department. 

Kind regards, 

on behalf of

Dr. Peter F. Biehl 

Academic Editor

PLOS ONE